# Escaping Saddle Points for Effective Generalization on Class-Imbalanced Data

**Harsh Rangwani**[*]    **Sumukh K Aithal**[*]    **Mayank Mishra**    **R. Venkatesh Babu**

Video Analytics Lab, Indian Institute of Science, Bengaluru, India

`{harshr@iisc.ac.in, sumukhaithal6@gmail.com,`
`mayankmishra@iisc.ac.in, venky@iisc.ac.in}`

## Abstract

Real-world datasets exhibit imbalances of varying types and degrees. Several techniques based on re-weighting and margin adjustment of loss are often used to enhance the performance of neural networks, particularly on minority classes. In this work, we analyze the class-imbalanced learning problem by examining the loss landscape of neural networks trained with re-weighting and margin based techniques. Specifically, we examine the spectral density of Hessian of class-wise loss, through which we observe that the network weights converges to a saddle point in the loss landscapes of minority classes. Following this observation, we also find that optimization methods designed to escape from saddle points can be effectively used to improve generalization on minority classes. We further theoretically and empirically demonstrate that Sharpness-Aware Minimization (SAM), a recent technique that encourages convergence to a flat minima, can be effectively used to escape saddle points for minority classes. Using SAM results in a 6.2% increase in accuracy on the minority classes over the state-of-the-art Vector Scaling Loss, leading to an overall average increase of 4% across imbalanced datasets. The code is available at https://github.com/val-iisc/Saddle-LongTail.

## 1 Introduction

In recent years, there has been a lot of progress in visual recognition thanks to the availability of well-curated datasets [30, 39], which are artificially balanced in terms of the frequency of samples across classes. However, modern real-world datasets are often imbalanced (*i.e.* long-tailed etc.) [29, 42, 43] and suffer from various kinds of distribution shifts. Overparameterized models like deep neural networks usually overfit classes with a high frequency of samples ignoring the minority (tail) ones [7, 43]. In such scenarios, when evaluated for metrics that focus on performance on minority data, these models perform poorly. These metrics are an essential and practical criterion for evaluating models in various domains like fairness [12], medical imaging [48] etc.

Many approaches designed for improving the generalization performance of models trained on imbalanced data, are based on the re-weighting of loss [14]. The relative weights for samples of each class are determined, such that the expected loss closely approximates the testing criterion objective [9]. In recent years, re-weighting techniques such as Deferred Re-Weighting (DRW) [9], and Vector Scaling (VS) Loss [28] have been introduced, which improve over the classical re-weighting method of weighting the loss of each class sample proportionally to the inverse of class frequency. However, even these improved re-weighting techniques lead to overfitting on the samples of tail classes. Also, it has been shown that use of re-weighted loss for training deep networks converges to final solutions similar to the un-weighted loss case, rendering it to be ineffective [8].

---

[*]Equal Contribution

36th Conference on Neural Information Processing Systems (NeurIPS 2022).

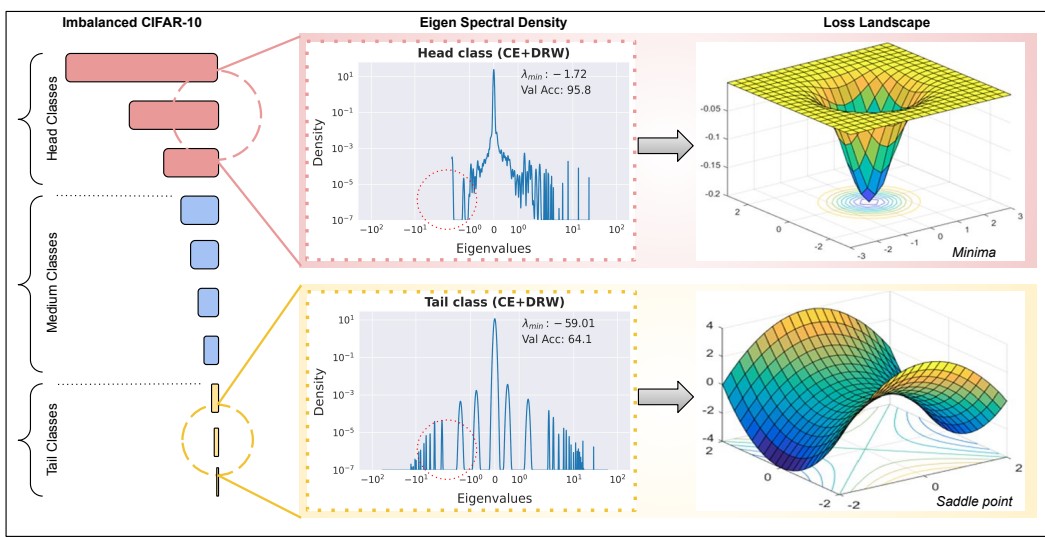

Figure 1: With class-wise Hessian analysis of loss, we observe that when deep neural networks are trained on class-imbalanced datasets, the final solution for tail classes reach a region of large negative curvature indicating convergence to saddle point (bottom), whereas the head classes converge to a minima (top). The properties of the loss landscape (saddle points or minima) can be observed by analyzing eigen spectral density (centre). [2]

This work looks at the loss landscape in weight space around final converged solutions for networks trained with re-weighted loss. We find that the generic Hessian-based analysis of the average loss used in prior works [19, 17], does not uncover any interesting insights about the sub optimal generalization on tail classes (Sec. 3). As the frequency of samples is different for each class due to imbalance, we analyze the Hessian of the loss for each class. This proposed way of analysis finds that re-weighting cannot prevent convergence to saddle points in the region of high negative curvature for tail classes, which eventually leads to poor generalization [16]. Whereas for head classes, the solutions converge to a minima with almost no significant presence of negative curvature, similar to networks trained on balanced data. This problem of converging to saddle points has not received much traction in recent times, as the negative eigenvalues disappear when trained on balanced datasets, indicating convergence to local minima [10, 19]. However, surprisingly our analysis shows that convergence to saddle points is still a practical problem for neural networks when they are trained on imbalanced (long-tailed) data (Fig. 1).

In the community, a lot of techniques optimization methods are designed to be able to escape saddle points efficiently [18, 24, 25], some of which involve adding a component of isotropic noise to gradients. However, these methods have not been able to improve the performance of deep networks in practice, as the implicit noise of SGD in itself mitigates the issue of saddle points when trained on balanced data [15, 25]. However in the case of imbalanced datasets, we find that the component of SGD along negative curvature (i.e., implicit noise) is insufficient to escape saddle points for minority classes. Thus, learning on imbalanced data can be serve as a practical benchmark for optimization algorithms that can escape saddle points.

We further demonstrate that Sharpness-Aware Minimization (SAM) [17] a recent optimization technique, with re-weighting can effectively enhance the gradient component along the negative curvature, allowing effective escape from saddle points which leads to improved generalization performance. We find that SAM can significantly improve the performance across various re-weighting and margin enhancing methods designed for long-tailed and class-imbalanced learning. The significant improvements are also observed on large-scale datasets of ImageNet-LT and iNaturalist 2018, demonstrating our resutls' applicability at scale. We summarize our contributions below:

- We propose class-wise Hessian analysis of loss which reveals convergence to saddle points in the loss landscape for minority classes. We find that even loss re-weighting solutions converge to saddle point, leading to sub-optimal generalization on the minority classes.

- We theoretically demostrate that SAM with re-weighting and high regularization factor significantly enhances the component of stochastic gradient along the direction of negative curvature , that results in effective escape from saddle points.
- We find that SAM can successfully enhance the performance of even state-of-the-art techniques for learning on imbalanced datasets which have a re-weighting component (*e.g.* VS Loss and LDAM) across various datasets and degrees of imbalance.

## 2 Related Work & Background

In this work, we use $g(x)$ to denote the output of a model, $\nabla g(x)$ to denote the gradients with respect to parameters, $x$ and $y$ denote the data and labels, respectively. We review the re-weighting methods used for training on imbalanced data with distribution shifts, followed by optimization techniques related to our work.

### 2.1 Long-Tailed Learning

Re-sampling [7] and Re-weighting [21] are the most commonly used methods to train on class-imbalanced datasets. Oversampling the minority classes [11] and undersampling the majority classes [7] are two approaches to re-sampling. Oversampling leads to overfitting on the tail classes, and undersampling discards a large amount of data, which inevitably results in poor generalization. Kang et al. [26] proposed to decouple representation learning and classifier training to improve performance with the same. Mixup Shifted Label-Aware Smoothing model (MiSLAS) [51] aims to improve the calibration of models trained on long-tailed datasets by mixup and label-aware smoothing and thereby improve performance. RIDE [45] and TADE [50] are ensemble-based methods that achieve state-of-the-art on the long-tailed visual recognition. Samuel and Chechik [41] introduces a new loss, DRO-LT, based on distributionally robust optimization for learning balanced feature representations. We explore the problem of training class-imbalanced datasets through the lens of optimization and loss landscape. We will now describe some representative recent effective methods in detail, which we will use as baselines. Additional discussion on long-tailed learning methods is present in App. H.

**LDAM [9]**: LDAM introduces optimal margins for each class based on reducing the error through a generalization bound. It results in the following loss function where $\Delta_j$ is the margin for each class:

$$\mathcal{L}_{\text{LDAM}}(y; g(x)) = -\log \frac{e^{g(x)_y - \Delta_y}}{e^{g(x)_y - \Delta_y} + \sum_{j \neq y} e^{g(x)_j}} \quad \text{where} \quad \Delta_j = \frac{C}{n_j^{1/4}} \text{ for } j \in \{1, \ldots, k\} \quad (1)$$

The core idea of LDAM is to regularize the classes with low frequency (low *i.e.* $n_j$) more, in comparison to the head classes with high frequency.

**DRW [9]**: Deferred Re- Weighting refers to training the model with average loss till certain epochs (K), then introducing weight $w_j$ proportional to $1/n_j$ to loss term specific to each class $j$ at a later stage. This way of re-weighting has been shown to be effective for improving generalization performance when combined with various losses such as Cross Entropy (CE), LDAM etc. We will be using CE+DRW method as a representative re-weighting method for our analysis. We define CE+DRW loss below for completeness:

$$\mathcal{L}_{\text{CE}}(y; g(x)) = -w_y \log(e^{g(x)_y} / \sum_{j=1}^{k} e^{g(x)_y}) \text{ where } w_j = \frac{1}{1 + (n_j - 1)\mathbb{1}_{epoch \geq K}} \quad (2)$$

**VS[28]**: Vector Scaling loss is a recently proposed loss function which unifies the idea of multiplicative shift (CDT shift [47]), additive shift (i.e Logit Adjustment [35]) and loss re-weighting. The final loss has the following form:

$$\mathcal{L}_{\text{VS}}(y; g(x)) = -w_y \log(e^{\gamma_y g(x)_y + \Delta_y} / \sum_{j=1}^{k} e^{\gamma_j g(x)_j + \Delta_j}) \quad (3)$$

Here the $\gamma_j$ and $\Delta_j$ are the multiplicative and additive logit hyperparameters, respectively.

---

[2]Figures for the minima and saddle point are from [4] and used for illustration purposes only.

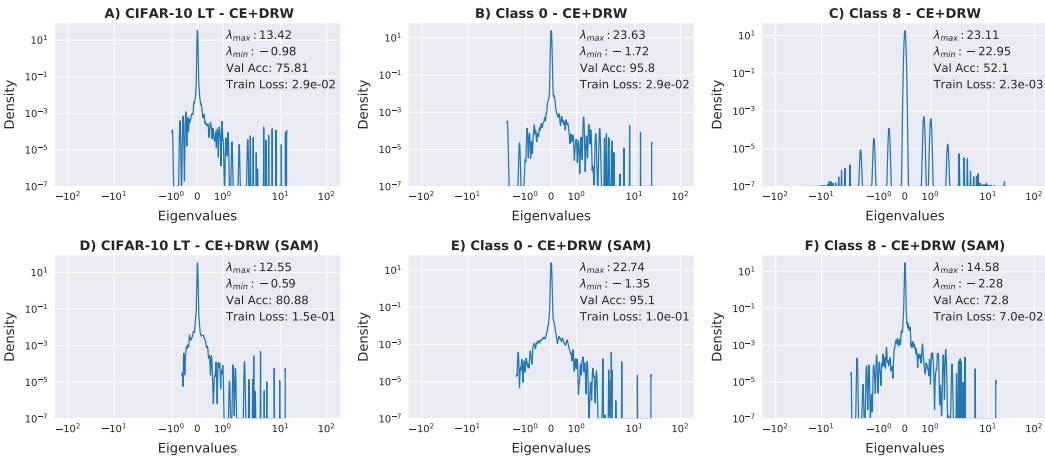

Figure 2: Eigen Spectral Density (Class-wise) on the head class (Class 0) and tail class (Class 8) with SGD and SAM. It can be observed that with the head classes, the validation accuracy with SGD (B) and SAM (E) are similar and the density of negative eigenvalues is not significant. On the tail class, SAM (F) escapes the saddle points (large density of negative eigenvalues) in SGD (C), leading to 20% increase in validation accuracy. A and D show the overall spectral density calculated across all samples in the dataset. Overall spectral density does not indicate the presence of saddles.

## 2.2 Loss Landscape

**Saddle Points**: Saddle points are regions in loss landscape that usually depict a plateau region with some negative curvature. In the non-convex setting, it has been shown that there is an existence of an exponential number of saddle points in loss landscape and convergence to these points demonstrate poor generalization [16]. There has been a lot of effort in developing methods for effectively escaping saddle points which involve the addition of noise (e.g., Perturbed Gradient Descent (PGD) [18, 24, 25]). However, these algorithms have not received much attention in the deep learning community as it has been shown that the implicit noise in SGD can escape saddles easily and converge to local minima [15]. Also, it has been empirically shown that negative eigenvalues from the Hessian spectrum disappear after a few steps of training, indicating escape from saddle points when neural networks are trained on balanced datasets [2, 10, 40]. However, contrary to this, we demonstrate that convergence to saddle points is prevalent in minority class loss landscapes and is a practical problem that can serve as a practical benchmark for the development of algorithms that escape saddle points.

**Flat Minima based Optimization methods**: Empirically, it has been shown that converging to a flat minima in loss landscape for a deep network leads to improved generalization in comparison to sharp minima [23, 27]. Recent works have tried to exploit this connection between the geometry of the loss landscape and generalization to achieve lower generalization error. Sharpness-Aware Minimization (SAM) [17] is one such algorithm that aims to simultaneously minimize the loss value and sharpness of the loss surface. SAM has shown impressive generalization abilities across various tasks including Natural Language processing [5], meta-learning [1] and domain adaptation [38]. Low-Pass Filtering SGD (LPF-SGD) [6] is another recently proposed optimization algorithm that aims to recover flat minima from the optimization landscape. LPF-SGD convolves the loss function with a Gaussian Kernel with variance proportional to the norm of the parameters of each filter in the network. In this work, we aim to explore the effectiveness of such algorithms for the task of escaping saddle points, which is a new direction for these algorithms.

## 3 Convergence to Saddle Points in Tail Class Loss Landscape

This section analyzes the dynamics of the loss landscape of neural networks trained on imbalanced datasets. We use the Cross Entropy (CE) loss $\hat{\mathcal{L}}_{\text{CE}}$ to denote the average cross entropy loss for each class. For fine-grained analysis, we focus on average loss on each class $\hat{\mathcal{L}}_{\text{CE}}(y)$. We visualize the loss landscape of the head and tail classes through the computation of Hessian Eigenvalue Density [19]. The Hessian of the train loss for each class $H = \nabla_w^2 \hat{\mathcal{L}}_{\text{CE}}(y)$ contains important properties about

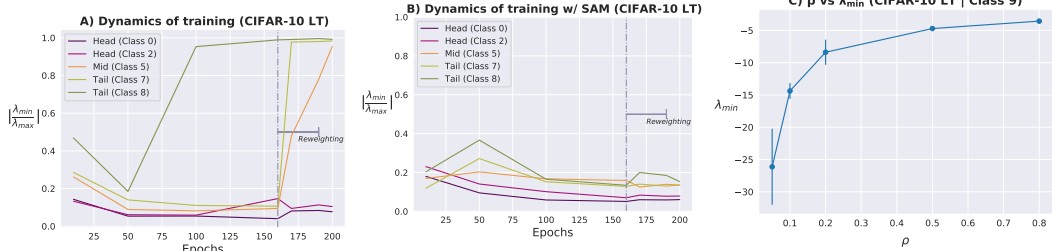

Figure 3: A) In CE+DRW, the tail class loss landscapes show significant non-convexity as indicated by the large value of $|\lambda_{min}/\lambda_{max}|$ whereas head classes (0,2) converge to convex landscapes. B) When CE+DRW is trained with SAM, it avoids convergence to non-convex regions throughout the training, as indicated by the low value of $|\lambda_{min}/\lambda_{max}|$. C) With high $\rho$, the $\lambda_{\min}$ increases, and the model converges to a point with low negative curvature (approx. minima).

the curvature of the classwise loss landscape. The Hessian Eigenvalue Density provides all suitable information regarding the eigenvalues of $H$. In this work, we focus on $\lambda_{max}$(max eigenvalue) and $\lambda_{min}$(min eigenvalue), which depict the extent of positive and negative curvature present. We use the Lanczos algorithm as introduced in Ghorbani et al. [19] to compute the Hessian Eigenvalue Density (spectral density) tractably. We further calculate the validation accuracy of a particular class $y$ and its eigen spectral density for analysis. We provide more details for these experiments in the App. D.

**Does the proposed class-wise analysis of loss landscape offer any additional insights?** In prior works [19, 20, 32], the Hessian of the average loss is used to characterize the nature of the converged point in the loss landscape. However, we find that when particularly trained on imbalanced datasets like CIFAR-10 LT, the eigen spectral density of the Hessian of average loss (Fig. 2A) does not differ from that of head class loss (Fig. 2B), indicating convergence to a local minima. However, explicitly analyzing the Hessian for the tail class loss (Fig. 2C) gives the correct indication of the presence of negative eigenvalues (i.e., curvature), which is in contrast to average loss. Hence, our proposed class-wise analysis of Hessian is essential for characterizing the nature of the converged solution when the training data is imbalanced.

**What happens when you train a neural network with CE-DRW method on CIFAR-10 LT?** Fig. 2 shows the spectral density on samples from the head class (Class 0 with 5000 samples) and tail class (Class 8 with 83 samples) at the checkpoint with the best validation accuracy. The spectral density of the head class contains few negative eigenvalues. Most of the eigenvalues are centered around zero, as also observed when training on a balanced dataset [19]. On the other hand, for the tail class, there exists a large number of both negative and positive eigenvalues, indicating convergence to a saddle point. We find that at this point, the $\hat{\mathcal{L}}_{CE}(y)$ is low along with the norm of gradient, which indicates a stationary saddle point. We also observe that the spectral density of the tail class contains many outlier eigenvalues, and $\lambda_{max}$ is much larger compared to the head class indicating sharp curvature. These evidences show that *the tail class solution converges to a saddle point instead of a local minimum*. Merkulov and Oseledets [36] indicated the existence of stationary points with low error but poor generalization in the loss landscape of neural networks. Also, the existence of saddle points being associated with poor generalization has been observed for small networks [16]. However, in this work, we show that convergence to saddle points can specifically occur in the loss landscape of tail classes even for the popular ResNet [22] family of networks, which is an important and novel observation to the best of our knowledge.

**Dynamics of training on Long-Tailed Datasets**: We analyze the $|\lambda_{min}/\lambda_{max}|$ for the head, mid and tail classes at various epochs (10, 50, 160, 170, 190, 200) across training to understand the dynamics of optimization with CE+DRW on long-tailed data (Fig. 3A). $|\lambda_{min}/\lambda_{max}|$ is a measure of non-convexity of the loss landscape [31], where a high value of $|\lambda_{min}/\lambda_{max}|$ conveys non-convexity indicating convergence to points with significant negative curvature. The network converges to non-convex regions with negative curvature for tail classes, showing convergence to the saddle point. Also, we find that for the certain tail (Class 7, 8) and mid classes (Class 5), the network starts converging towards regions with negative curvature after applying loss re-weighting (DRW at 160th epoch). This indicates that DRW leads to convergence to a saddle point rather than preventing it.

# 4 Escaping Saddle Points for Improved Generalization

In this section, we analyze the Sharpness-Aware Minimization technique for escaping from saddle points in tail class loss landscape. In existing works [3, 33, 52], the effectiveness of SAM in escaping saddle points has not been explored to the best of our knowledge.

**Sharpness-Aware Minimization (SAM)**: Sharpness-Aware Minimization is a recent technique which aims to flatten the loss landscape by first finding a sharp maximal point $\epsilon$ in the neighborhood of current weight $w$. It then minimizes the loss at the sharp point $(w + \epsilon)$.

$$\min_w \max_{||\epsilon|| \leq \rho} f(w + \epsilon) \tag{4}$$

Here, $f$ is any objective (eg. CE or LDAM loss function) and $\rho$ is the hyperparameter that controls the extent of neighborhood. A high value of $\rho$ leads to convergence to much flat loss landscape. The inner optimization in above objective is first approximated using a first order solution:

$$\hat{\epsilon}(w) \approx \arg\max_{||\epsilon|| \leq \rho} f(w) + \epsilon^T \nabla f(w) \ = \rho \nabla f(w) / ||\nabla f(w)||_2 \tag{5}$$

After finding $\hat{\epsilon}(w)$, the network weights are updated using the gradient $\nabla f(w)|_{w+\hat{\epsilon}(w)}$. In recent work [3], it has been shown that the normalization of the norm of the gradient for $\hat{\epsilon}(w)$ calculation above leads to oscillation which implies non-convergence theoretically. Also, it has been empirically shown that the unnormalized version of the gradient with adjusted $\rho$ performs better than the normalized version. Hence, we use the approximation i.e. $\hat{\epsilon}(w) = \rho \nabla f(w)$ for our theoretical results. As we will be using the stochastic version of the gradient, we use $z$ as the stochasticity parameter and denote the gradient as $\nabla f_z(w)$. With this, we now define the gradient with respect to $w$ that is associated with SAM:

$$\nabla f_z^{\text{SAM}}(w) = \nabla f_z(w + \hat{\epsilon}(w)) = \nabla f_z(w + \rho \nabla f_z(w)) \tag{6}$$

As we are using the same batch for obtaining the gradient to calculate the $\hat{\epsilon}(w)$ and loss, we can use the same $z$ as the argument. We now analyze the component of the SAM gradient in the direction of negative curvature, which is required for escaping saddle points [15].

## 4.1 Analysis of SAM for Escaping Saddle Points

Our analysis is based on the Correlated Negative Curvature (CNC) assumption [15] that states that stochastic gradients have components along the direction of negative curvature, which helps them escape from saddle points. This assumption has been shown to be theoretically valid for the problem of learning half-spaces and also has been empirically verified for a large number of neural networks of different sizes [15]. We now formally state the assumption below:

**ASSUMPTION 1** (Correlated Negative Curvature [15]). Let $\mathbf{v_w}$ be the minimum eigenvector corresponding to the minimum eigenvalue of the Hessian matrix $\nabla^2 f(w)$. The stochastic gradient $\nabla f_{\mathbf{z}}(w)$ satisfies the CNC assumption if the second moment of the projection along the direction $\mathbf{v_w}$ is uniformly bounded away from zero, i.e.

$$\exists \, \gamma \geq 0 \; s.t. \; \forall w : \mathbf{E}[< \mathbf{v_w}, \nabla f_{\mathbf{z}}(w) >^2] \geq \gamma \tag{7}$$

It has also been emphasized that the value of $\gamma$ is shown to correlate with the magnitude of $\lambda_{\min}^2$. This shows that with a high negative eigenvalue, there is a large component of gradient along the negative curvature along $\boldsymbol{v_w}$. This allows the SGD algorithms to escape the saddle points. However, we find that in the case of class imbalanced learning (Fig. 1) even stochastic gradients may have an insufficient component in the direction of negative curvature to escape the saddle points. We now show that SAM technique, which aims to reach a flat minima, further amplifies the gradient component along negative curvature and can be effectively used to escape the saddle point. We now formally state our theorem based on the CNC assumption below:

**THEOREM 2.** *Let $\mathbf{v_w}$ be the minimum eigenvector corresponding to the minimum eigenvalue $\lambda_{\min}$ of the Hessian matrix $\nabla^2 f(w)$. The $\nabla f_{\mathbf{z}}^{SAM}(w)$ satisfies that it's second moment of projection in $v_w$ is atleast $(1 + \rho \lambda_{\min})^2$ times the original (component of $\nabla f_{\mathbf{z}}(w)$):*

$$\exists \, \gamma \geq 0 \; s.t. \; \forall w : \mathbf{E}[< \mathbf{v_w}, \nabla f_{\mathbf{z}}^{SAM}(w) >^2] \geq (1 + \rho \lambda_{min})^2 \gamma \tag{8}$$

Table 1: Results on CIFAR-10 LT and CIFAR-100 LT with $\beta$=100. SAM with re-weighting is able to avoid the regions of negative curvature, leading to major gain in performance on the mid and tail classes with CE, LDAM and VS.

| | CIFAR-10 LT | | | | CIFAR-100 LT | | | |
|---|---|---|---|---|---|---|---|---|
| | Acc | Head | Mid | Tail | Acc | Head | Mid | Tail |
| CE | $71.7_{\pm0.1}$ | $90.8_{\pm3.6}$ | $71.9_{\pm0.4}$ | $52.3_{\pm3.7}$ | $38.5_{\pm0.5}$ | $64.5_{\pm0.7}$ | $36.8_{\pm1.0}$ | $8.2_{\pm1.0}$ |
| CE + SAM | $73.1_{\pm0.3}$ | $93.3_{\pm0.2}$ | $74.1_{\pm0.6}$ | $51.7_{\pm1.0}$ | $39.6_{\pm0.6}$ | $66.5_{\pm0.7}$ | $38.1_{\pm1.1}$ | $8.0_{\pm0.6}$ |
| CE + DRW [9] | $75.5_{\pm0.2}$ | $91.6_{\pm0.4}$ | $74.1_{\pm0.4}$ | $61.4_{\pm0.9}$ | $41.0_{\pm0.6}$ | $61.3_{\pm1.3}$ | $41.7_{\pm0.5}$ | $14.7_{\pm0.9}$ |
| CE + DRW + SAM | $80.6_{\pm0.4}$ | $91.4_{\pm0.3}$ | $78.0_{\pm0.4}$ | $73.1_{\pm0.9}$ | $44.6_{\pm0.4}$ | $61.2_{\pm0.8}$ | $47.5_{\pm0.6}$ | $20.7_{\pm0.6}$ |
| LDAM + DRW [9] | $77.5_{\pm0.5}$ | $91.1_{\pm0.8}$ | $75.7_{\pm0.7}$ | $66.4_{\pm0.2}$ | $42.7_{\pm0.3}$ | $61.8_{\pm0.6}$ | $42.2_{\pm1.5}$ | $19.4_{\pm0.9}$ |
| LDAM + DRW + SAM | $81.9_{\pm0.4}$ | $91.0_{\pm0.2}$ | $79.2_{\pm0.5}$ | $76.4_{\pm1.1}$ | $45.4_{\pm0.1}$ | $64.4_{\pm0.3}$ | $46.2_{\pm0.2}$ | $20.8_{\pm0.3}$ |
| VS [28] | $78.6_{\pm0.3}$ | $90.6_{\pm0.4}$ | $75.8_{\pm0.5}$ | $70.3_{\pm0.5}$ | $41.7_{\pm0.5}$ | $54.4_{\pm0.2}$ | $41.1_{\pm0.6}$ | $26.8_{\pm1.0}$ |
| VS + SAM | $82.4_{\pm0.4}$ | $90.7_{\pm0.0}$ | $79.6_{\pm0.5}$ | $78.0_{\pm01.2}$ | $46.6_{\pm0.4}$ | $56.4_{\pm0.4}$ | $48.8_{\pm0.6}$ | $31.7_{\pm0.1}$ |

**REMARK.** The above theorem adds the factor $(1 + \rho\lambda_{\min})^2$ to increase the component in direction of negative curvature ($\gamma$) when $\lambda_{\min} \leq \frac{-2}{\rho}$. Due to this increase, the model will be able to escape from directions with high negative curvature, leading to an increased $\lambda_{\min}$. Also, as the factor $\frac{-2}{\rho}$ is inversely proportional to $\rho$, the high value of $\rho$ aids in effectively increasing the minimum negative eigenvalue. To empirically verify this, we evaluate the Hessian spectrum for the CIFAR-10 LT dataset using CE-DRW method for different values of $\rho$ (Fig. 3C). We find that, as expected from the theorem, in practice, the high values of $\rho$ lead to less negative values of $\lambda_{\min}$. This indicates escaping the saddle points effectively, hence avoiding convergence to regions having negative curvature in loss landscape. The proof of the above theorem and additional details is provided in Appendix B.

We also want to convey that theoretically, techniques like Perturbed Gradient Descent (PGD), and LPF-SGD (Low-Pass Filter SGD), which add Gaussian noise into gradient to escape saddle points can also be used for mitigating negative curvature. Also it has been found that SGD [15] can also escape the saddle points and converges to solutions with a flat loss landscape. Also, theoretically according to Theorem 2 in Daneshmand et al. [15] the SGD algorithm convergence to a second-order stationary point depends on the $\gamma$ as $\mathcal{O}(\gamma^{-4})$ under some assumptions on $f$. As we find that as SAM with high $\rho$ enhances the component of SGD in direction of negative curvature ($\gamma$) by $(1+\rho\lambda_{\min})^2$, it is reasonable to expect that SAM is able to escape saddle points effectively and converge to solutions with significant less negative curvature quickly implying better generalization. We provide empirical evidence for this in Fig. 3B and Sec. 5.2.

**What happens when you train a neural network with SAM + DRW?** With SAM (high $\rho$), the large negative eigenvalues present in the loss landscape of the tail class get suppressed (Fig. 2F). In the spectral density for the tail class, it can be seen that $\lambda_{min}$ is much closer to zero for SAM compared to its counterpart with SGD. This aligns with the hypothesis that SAM escapes regions of negative curvature, leading to improved accuracy on the tail classes. However, the spectral density of the head class does not change significantly compared to that of Empirical Risk Minimization (ERM), although the $\lambda_{max}$ is much lower for SAM, indicating a flatter minima for the head class.

We also analyze the $|\lambda_{min}/\lambda_{max}|$ across multiple steps of training with SAM (Fig. 3B), where $|\lambda_{min}/\lambda_{max}|$ is a measure of non-convexity of the loss surface. We observe that SAM does not allow the tail classes to reach a region of high non-convexity. The values of $|\lambda_{min}/\lambda_{max}|$ is much lower for SAM compared to SGD (Fig. 3A) throughout training, indicating minimal negative eigenvalues (*i.e.* more convexity) in the loss landscape, especially for the tail and medium classes. This clearly shows that SAM avoids regions of substantial negative curvature in the search of flat minima. Further, we note that once the re-weighting begins, SAM is able to avoid convergence to a saddle point (non-convexity decreases), which is contrary to what we observe with CE+DRW (with SGD). Theorem 2 states that SAM consists of larger component in the direction of negative curvature which allows to reach a solution with minimal negative curvature. Empirically, Fig. 3B also supports the Theorem 2 as we observe that SAM reaches a minima (high convexity) for all the classes.

Table 2: Results on CIFAR-10 and CIFAR-100 with Step Imbalance ($\beta = 100$). SAM generalizes well across datasets with varied type of imbalance, resulting in substantial gain in tail accuracy in all settings.

| | CIFAR-10 | | | CIFAR-100 | | |
|---|---|---|---|---|---|---|
| | Acc | Head | Tail | Acc | Head | Tail |
| CE | 65.1 | 88.6 | 41.7 | 38.6 | 76.3 | 00.9 |
| CE + SAM | 66.1 | 92.9 | 39.4 | 39.3 | 78.6 | 00.0 |
| CE + DRW [9] | 72.2 | 93.1 | 51.2 | 45.8 | 73.9 | 17.8 |
| CE + DRW + SAM | 79.3 | 92.7 | 65.8 | 48.3 | 73.1 | 23.4 |
| LDAM + DRW [9] | 77.6 | 89.2 | 66.0 | 45.3 | 70.3 | 20.4 |
| LDAM + DRW + SAM | 81.0 | 90.5 | 71.5 | 49.2 | 74.0 | 24.4 |
| VS [28] | 77.0 | 91.7 | 62.3 | 46.5 | 69.0 | 24.1 |
| VS + SAM | 82.0 | 91.7 | 72.3 | 48.3 | 70.4 | 26.2 |

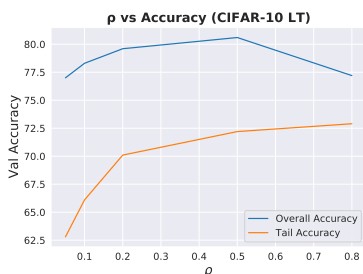

Figure 4: Impact of $\rho$ (regularization factor) on Overall Accuracy and Tail Accuracy (CIFAR-10 LT).

## 5 Experiments

### 5.1 Class-Imbalanced Learning

**Datasets**: We report our results on four long-tailed datasets: CIFAR-10 LT [9], CIFAR-100 LT [9], ImageNet-LT [34], and iNaturalist 2018 [44]. **a) CIFAR-10 LT and CIFAR-100 LT**: The original CIFAR-10 and CIFAR-100 datasets consist of 50,000 training images and 10,000 validation images, spread across 10 and 100 classes, respectively. We use two imbalance versions, i.e., long-tail imbalance and step imbalance, as followed in Cao et al. [9]. The imbalance factor, $\beta = \frac{N_{\max}}{N_{\min}}$, denotes the ratio between the number of samples in the most frequent ($N_{\max}$) and least frequent class ($N_{\min}$). For both the imbalanced versions, we analyze the results with $\beta = 100$. **b) ImageNet-LT and iNaturalist 2018**: We use the ImageNet-LT version as proposed by [34], which is an class-imbalanced version of the large-scale ImageNet dataset [39]. It consists of 115.8K images from 1000 classes, with 1280 images in the most frequent class and 5 images in the least. iNaturalist 2018 [44] is a real-world long-tailed dataset that contains 437.5K images from 8,142 categories. In the case of long-tail imbalance, we segregate the classes of all the datasets into *Head* (Many), *Mid* (Medium), and *Tail* (Few) subcategories, as defined in [51]. For step imbalance experiments on CIFAR datasets, we split the classes into *Head* (Frequent) and *Tail* (Minority), as done in [9].

**Experimental Details**: We follow the hyperparameters and setup as in Cao et al. [9] for CIFAR-10 LT and CIFAR-100 LT datasets. We train a ResNet-32 architecture as the backbone and SGD with a momentum of 0.9 as the base optimizer for 200 epochs. A multi-step learning rate schedule is used, which drops the learning rate by 0.01 and 0.0001 at the 160th and 180th epoch, respectively. For training with SAM, we set a constant $\rho$ value of either 0.5 or 0.8 for most methods. For ImageNet-LT and iNaturalist 2018 datasets, we use the ResNet-50 backbone similar to [51]. An initial learning rate of 0.1 and 0.2 is set for iNaturalist 2018 and ImageNet-LT, respectively, followed by a cosine learning rate schedule. We initialize the $\rho$ value with 0.05 and utilize a step schedule to increase the $\rho$ value during the course of training for SAM experiments. We run every experiment on long-tailed CIFAR datasets with three seeds and report the mean and standard deviation. Additional implementation details are provided in the App. C. Algorithm for DRW+SAM is defined in App. G.

**Baselines**: **a) Cross-Entropy (CE)**: CE minimizes the average loss across all samples, and thus, the performance of tail classes is much lower than that of head classes. **b) CE + Deferred Re-Weighting (DRW) [9]**: The re-weighting of CE loss inversely by class frequency is done in the later stage of training. **c) LDAM + DRW [9]**: Label-Distribution-Aware Margin (LDAM) proposes a margin-based loss that encourages larger margins for less-frequent classes. **d) Vector Scaling (VS) Loss [28]**: VS loss incorporates both additive and multiplicative logit adjustments to modify inter-class margins.

**Results**: Table 1 summarizes our results on CIFAR-10 LT and CIFAR-100 LT with $\beta$ of 100. It can be observed that SAM with re-weighting significantly improves the accuracy on mid and tail classes while preserving the accuracy on head classes. SAM improves upon the overall performance

Table 3: Results on iNaturalist 2018 and ImageNet-LT datasets with LDAM+DRW and comparison with other methods. The numbers for methods marked with † are taken from [51].

| Method | Two stage | iNaturalist 2018 | | | | ImageNet-LT | | | |
|---|---|---|---|---|---|---|---|---|---|
| | | Acc | Head | Mid | Tail | Acc | Head | Mid | Tail |
| CE | ✗ | 60.3 | 72.8 | 62.7 | 54.8 | 42.7 | 62.5 | 36.6 | 12.5 |
| cRT [26] † | ✓ | 68.2 | 73.2 | 68.8 | 66.1 | 50.3 | 62.5 | 47.4 | 29.5 |
| LWS [26] † | ✓ | 69.5 | 71.0 | 69.8 | 68.8 | 51.2 | 61.8 | 48.6 | 33.5 |
| MiSLAS [51] | ✓ | **71.6** | 73.2 | **72.4** | 70.4 | 52.7 | 61.7 | 51.3 | **35.8** |
| DisAlign [49] | ✓ | 69.5 | 61.6 | 70.8 | 69.9 | 52.9 | 61.3 | **52.2** | 31.4 |
| DRO-LT [41] | ✗ | 69.7 | **73.9** | 70.6 | 68.9 | **53.5** | **64.0** | 49.8 | 33.1 |
| CE + DRW | ✗ | 63.0 | 59.8 | 64.4 | 62.3 | 44.9 | 57.9 | 42.2 | 21.6 |
| CE + DRW + SAM | ✗ | 65.3 | 60.5 | 66.2 | 65.5 | 47.1 | 56.6 | 45.8 | 28.1 |
| LDAM + DRW | ✗ | 67.5 | 63.0 | 68.3 | 67.8 | 49.9 | 61.1 | 48.2 | 28.3 |
| LDAM + DRW + SAM | ✗ | 70.1 | 64.1 | 70.5 | **71.2** | 53.1 | 62.0 | 52.1 | 34.8 |

of CE+DRW by 5.1% on CIFAR-10 LT and 3.6% on CIFAR-100 LT datasets, with the tail class accuracy increasing by 11.7% and 7.7% respectively. These results empirically show that escaping saddle points with SAM leads to a notable increase in overall accuracy primarily due to the major gain in the accuracy on the tail classes. The addition of SAM to recently proposed long-tail learning methods like LDAM and VS loss leads to a significant increase in performance, which indicates that the role of SAM is orthogonal to the margin-based methods. On the other hand, SAM without re-weighting (CE+SAM) improves accuracy on the head and mid classes rather than the tail class. This can be attributed to the fact that standard ERM minimizes the average loss across all the samples without re-weighting such that the weightage of tail class samples in the overall loss is minimal. This shows that naive application of SAM is ineffective in improving tail class performance, in comparison to proposed combination of re-weighting methods with SAM. We also show improved results with various imbalance factors ($\beta$) in App. F.

We also show results with step imbalance ($\beta = 100$) on CIFAR-10 and CIFAR-100 datasets (Table 2). With step imbalance on CIFAR-10, the first five classes have 5000 samples each, and the remaining classes have 50 samples each. The addition of SAM improves the overall performance of CE+DRW on CIFAR-10 by 7.1%, with the tail class accuracy increasing by 14.6%. We observe that on most tail classes, the density of negative eigenvalues in the spectral density is much lower with SAM. This indicates that despite multiple classes with few samples, SAM with DRW can avoid the saddle points. SAM systematically improves performance with LDAM and VS loss leading to state-of-the-art performance on both CIFAR-10 and CIFAR-100 in the step imbalance setting.

**Do these observations scale to large-scale datasets?** We report the results on ImageNet-LT dataset in Table 3. We also compare with recent long-tail learning methods: cRT [26], MisLAS [51], DisAlign [49] and DRO-LT [41]. The observations on CIFAR-10 LT and CIFAR-100 LT hold good even on ImageNet-LT. For example, the accuracy on tail classes increases by 6.5% with the introduction of SAM on CE + DRW, which is similar to the gain observed in CIFAR-100 LT with CE + DRW. We observe that LDAM+DRW+SAM surpasses the performance of two-stage training methods including MisLAS, cRT, LWS, and DisAlign. Compared to these two-stage methods, our method is a single stage method and outperforms these two-stage methods. These observations point out that the problem of saddle points also exists in large datasets and convey that SAM is easily generalizable to large-scale imbalanced datasets without making any significant changes. On iNaturalist 2018 [44] too, the accuracy on tail classes gets boosted by more than 3% with SAM (Table 3).

**Comparison with SOTA**: VS loss [28] is a recently proposed margin-based method that achieves state-of-the-art performance on class-imbalanced datasets with single-stage training without strong augmentations [51], ensembles [50] or self-supervision [46]. SAM significantly improves upon the performance of VS on both CIFAR-10 LT and CIFAR-100 LT. For the practitioners, we suggest *using high $\rho$ SAM with re-weighting or margin based methods* for effective learning on long-tailed data. We also integrate SAM with more recent IB-Loss [37] and Parametric Contrastive Learning (PaCo) [13] methods and report the results in App. E. We find that SAM is also effectively able to improve performance of these recent methods.

Table 4: Results on CIFAR-10 LT and CIFAR-100 LT with various methods that escape saddle points.

| | CIFAR-10 LT | | | | CIFAR-100 LT | | | |
|---|---|---|---|---|---|---|---|---|
| | Acc | Head | Mid | Tail | Acc | Head | Mid | Tail |
| CE + DRW | 75.5 | 91.6 | 74.1 | 61.4 | 41.0 | 61.3 | 41.7 | 14.7 |
| CE + DRW + PGD [24] | 77.2 | 92.0 | 75.2 | 65.0 | 42.2 | 63.0 | 41.6 | 17.0 |
| CE + DRW + LPF-SGD [6] | 78.5 | 90.8 | 77.7 | 67.2 | 42.9 | 64.0 | 43.7 | 15.8 |
| CE + DRW + SAM | 80.6 | 91.4 | 78.0 | 73.1 | 44.6 | 61.2 | 47.5 | 20.7 |

## 5.2 Ablation Studies

**A note on $\rho$ value**: We observe that as we increase the smoothness parameter ($\rho$) in SAM, the accuracy on the tail classes increases significantly (Fig. 4). The accuracy on tail classes increases from 63% for $\rho = 0.05$ to 73% for $\rho = 0.8$ on CIFAR-10 LT with CE+DRW. This can be ascribed to the correlation between $\lambda_{min}$ and $\rho$ as discussed in Sec. 4.1. As the $\rho$ increases, the negative curvature in the tail classes disappears because SAM aims to find a flat minima with a large neighborhood with a low loss value. A very large $\rho$ (0.8) leads to a drop in the head accuracy because it restricts the solution space of the head class, resulting in a drop in the overall accuracy. This also emphasizes that a high $\rho$ is necessary for escaping saddle points and achieving the best results.

**Other methods to escape saddle points**: In Table 4, we show that other methods developed to escape saddle points, such as PGD, can be used for improving generalization on tail classes. LPF-SGD, an algorithm promoting convergence to flat landscape, inherently adds Gaussian noise to the network parameters and could be considered similar to PGD. We can see that the addition of PGD and LPF-SGD to CE+DRW leads to a substantial gain in the performance of tail classes on CIFAR-10 LT and CIFAR-100 LT. It can also be observed that CE+DRW+SAM outperforms both PGD and LPF-SGD by 2% on average. This further highlights that various methods in literature developed to escape saddle points efficiently can be directly used to improve the performance of minority classes when training on class-imbalanced datasets.

## 6 Conclusion

In this work, we show that training on imbalanced datasets can lead to convergence to points with sufficiently large negative curvature in the loss landscape for minority classes. We find that this is quite common when neural networks are trained with loss functions that are re-weighted or modified to enhance the focus on minority classes. Due to the occurrence of saddle points, we observe that the network suffers from poor generalization on minority classes. We propose to use Sharpness-Aware Minimization (SAM) with a high regularization factor $\rho$ as an effective method to escape regions of negative curvature and enhance the generalization performance. We theoretically and empirically demonstrate that SAM with high $\rho$ is able to escape saddle points faster than SGD and converge to better solutions, which is a novel observation to the best of our knowledge. We show that combining SAM with state-of-the-art techniques for learning with imbalanced data leads to significant gains in performance on minority classes. We hope that our work leads to further research in studying the effect of negative curvature in generalization as we show they are a practical issue for class-imbalanced learning using deep neural networks.

**Acknowledgements**: This work was supported in part by SERB-STAR Project (Project:STR/2020/000128), Govt. of India. Harsh Rangwani is supported by Prime Minister's Research Fellowship (PMRF). We are thankful for their support.

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
