# Supplementary
# Escaping Saddle Points for Effective Generalization on Class-Imbalanced Data

Harsh Rangwani*     Sumukh K Aithal*     Mayank Mishra     R. Venkatesh Babu
Video Analytics Lab, Indian Institute of Science, Bengaluru, India
{harshr@iisc.ac.in, sumukhaithal6@gmail.com,
mayankmishra@iisc.ac.in, venky@iisc.ac.in}

## Contents

## A  Limitations of Our Work

We would like to highlight that our theoretical results are based on Daneshmand et al. [6] which verified CNC condition for small scale neural networks, verifying the CNC condition for large networks and exactly characterizing the saddle point solutions obtained by SAM for minority classes, are good directions for future work.

Also empirically, we propose to use Sharpness-Aware Minimization with high $\rho$ for tail classes to escape from saddle points. Although the general guideline is to use a higher $\rho$ value like 0.5 or 0.8 to achieve the best result, we do find that $\rho$ as a hyperparameter still requires tuning to obtain the best results. We believe making SAM hyper-parameter free is an interesting direction to pursue in the future.

---

*Equal Contribution

36th Conference on Neural Information Processing Systems (NeurIPS 2022).

# B  Proof of Theorem

In this section, we re-state Theorem 2 and provide it's proof. The theorem analyzes the variance of stochastic gradient for SAM along the direction of negative curvature and shows that SAM amplifies the variance by a factor, which signals that it has a stronger component in direction of negative curvature under certain conditions. Hence, SAM can be used for effectively escaping saddle points in the loss landscape. This is based on Correlated Negative Curvature (CNC) Assumption for stochastic gradients (Assumption 1). The $\mathbf{v_w}, \nabla f(w) \in \mathbb{R}^{p \times 1}$ whereas the Hessian denoted by $H(f(w))$ (also $\nabla^2 f(w)) \in \mathbb{R}^{p \times p}$ where $p$ is the number of parameters in the model.

**THEOREM 1.** *Let $\mathbf{v_w}$ be the minimum eigenvector corresponding to the minimum eigenvalue $\lambda_{\min}$ of the Hessian matrix $\nabla^2 f(w)$. The $\nabla f_{\mathbf{z}}^{SAM}(w)$ satisfies that it's second moment of projection in $v_w$ is atleast $(1 + \rho\lambda_{min})^2$ times the original (component of $\nabla f_{\mathbf{z}}(w)$):*

$$\exists\, \gamma \geq 0 \; s.t. \; \forall w : \mathbf{E}[< \mathbf{v_w}, \nabla f_{\mathbf{z}}^{SAM}(w) >^2] \geq (1 + \rho\lambda_{min})^2 \gamma \tag{1}$$

*Proof.* Using the first-order approximation of a vector valued function through Taylor series:

$$f(w + \epsilon) = f(w) + J(\nabla f(w))\epsilon \tag{2}$$

here $J$ is the jacobian operator. After considering $\rho$ to be small we have the following approximation for the SAM gradient:

$$\nabla f^{SAM}(\boldsymbol{w}) = \nabla f(w + \rho\nabla f(w)) \tag{3}$$
$$= \nabla f(w) + \rho H(f(w))\nabla f(w) \tag{4}$$

Here, we have used the following property that $J(\nabla f(w))$ is the Hessian matrix $H(f(w))$ (also written as $\nabla^2 f(w)$). Also, as we now want to work with stochastic gradients, we replace gradient $\nabla f(w)$ with it's stochastic version $\nabla f_z(w)$ and introduce an expectation expression. Now, we analyze the second-moment of the SAM gradient along the direction of most negative curvature $\boldsymbol{v_w}$:

$$\begin{aligned}
\boldsymbol{E}[< \mathbf{v_w}, \nabla f_{\mathbf{z}}^{SAM}(w) >^2] &= \boldsymbol{E}[< \mathbf{v_w}, \nabla f_z(w) + \rho H(f(w))\nabla f_z(w)) >^2] \\
&= \boldsymbol{E}[(< \mathbf{v_w}, \nabla f_z(w) > + \rho < \mathbf{v_w}, H(f(w))\nabla f_z(w) >)^2] \\
&= \boldsymbol{E}[(< \mathbf{v_w}, \nabla f_z(w) > + \rho \mathbf{v_w}^T H(f(w))\nabla f_z(w))^2]
\end{aligned}$$

Here, we use the matrix notation for dot product $< x, y > = x^T y$. Using the property of the eigen vector: $\mathbf{v_w^T} H(f(w)) = \lambda_{min}\mathbf{v_w^T}$, we substitute the value below:

$$\begin{aligned}
\boldsymbol{E}[< \mathbf{v_w}, \nabla f_{\mathbf{z}}^{SAM}(w) >^2] &= \boldsymbol{E}[(< \mathbf{v_w}, \nabla f_z(w) > + \rho\lambda_{min}\mathbf{v_w^T}\nabla f_z(w) >)^2] \\
&= \boldsymbol{E}[(< \mathbf{v_w}, \nabla f_z(w) > + \rho\lambda_{min} < \mathbf{v_w}, \nabla f_z(w) >)^2] \\
&= \boldsymbol{E}[((1 + \rho\lambda_{min}) < \mathbf{v_w}, \nabla f_z(w) >)^2] \\
&= (1 + \rho\lambda_{min})^2 \boldsymbol{E}[< \mathbf{v_w}, \nabla f_z(w) >^2] \\
&\geq (1 + \rho\lambda_{min})^2 \gamma
\end{aligned}$$

The last step follows from the CNC Assumption 1. This completes the proof. ∎

# C  Experimental Details

**Imbalanced CIFAR-10 and CIFAR-100**: For the long-tailed imbalance (CIFAR-10 LT and CIFAR-100 LT), the sample size across classes decays exponentially with $\beta = 100$. CIFAR-10 LT holds 5000 samples in the most frequent class and 50 in the least, whereas CIFAR-100 LT decays from 500 samples in the most frequent class to 5 in the least. The classes are divided into three subcategories: *Head* (Many), *Mid* (Medium), and *Tail* (Few). For CIFAR-10 LT, the first 3 classes (> 1500 images each) fall into the head classes, following 4 classes (> 250 images each) into the mid classes, and the final 3 classes (< 250 images each) into the tail classes. Whereas for CIFAR-100 LT, head classes consist of the initial 36 classes, mid classes contain the following 35 classes, and the tail classes consist of the remaining 29 classes.

Table 1: $\rho$ value for used for reporting the results with SAM.

| | CIFAR-10 | | CIFAR-100 | |
| --- | --- | --- | --- | --- |
| | LT ($\beta = 100$) | Step ($\beta = 100$) | LT ($\beta = 100$) | Step ($\beta = 100$) |
| CE + SAM | 0.1 | 0.1 | 0.2 | 0.5 |
| CE + DRW + SAM | 0.5 | 0.2 | 0.8 | 0.2 |
| LDAM + DRW + SAM | 0.8 | 0.1 | 0.8 | 0.5 |
| VS + SAM | 0.5 | 0.2 | 0.8 | 0.2 |

In the step imbalance setting, both CIFAR-10 and CIFAR-100 are split into two classes, i.e., *Head* (Frequent) and *Tail* (Minority), with $\beta = 100$. The first 5 (Head) classes of CIFAR-10 contain 5000 samples each, along with 50 samples each in the remaining 5 (Tail) classes. On the other hand, the top first 50 (Head) classes of CIFAR-100 contain 500 samples each, and the remaining 50 (Tail) classes consist of 5 samples each.

All the experiments on imbalanced CIFAR-10 and CIFAR-100 are run with ResNet-32 backbone and SGD with momentum 0.9 as the base optimizer. All the methods train on imbalanced CIFAR-10 and CIFAR-100 with a batch size of 128 for 200 epochs, except for VS Loss, which runs for 300 epochs. We follow the learning rate schedule mentioned in Cao et al. [3]. In the initial 5 epochs, we linearly increase the learning rate to reach 0.1. Following that, a multi-step learning rate schedule decays the learning rate by scaling it with 0.001 and 0.0001 at 160th and 180th epoch, respectively. For LDAM runs on imbalanced CIFAR, the value of $C$ is tuned so that $\Delta_j$ is normalised to set maximum margin of 0.5 (refer to Equation. 1 in main text). In the case of VS Loss, we use $\gamma$ as 0.05 and $\tau$ as 0.75 for imbalanced CIFAR-10 and CIFAR-100 datasets (refer to Equation. 3 in main text).

**ImageNet-LT and iNaturalist 2018**: The classes in ImageNet-LT and iNaturalist 2018 datasets are also divided into three subcategories, i.e., *Head* (Many), *Mid* (Medium), and *Tail* (Few). For ImageNet-LT, the head classes consist of the first 390 classes, mid classes contain the subsequent 445 classes, and the tail classes hold the remaining 165 classes. Whereas for iNaturalist 2018, first 842 classes fall into the head classes, subsequent 3701 classes into the mid classes, and the remaining 3599 into the tail classes.

For ImageNet-LT and iNaturalist 2018, all the models are trained for 90 epochs with a batch size of 256. We use ResNet-50 architecture as the backbone and SGD with momentum 0.9 as the base optimizer. A cosine learning rate schedule is deployed with an initial learning rate of 0.1 and 0.2 for iNaturalist 2018 and ImageNet-LT, respectively. For LDAM runs on ImageNet-LT and iNaturalist 2018, the value of $C$ is tuned so that $\Delta_j$ is normalised to set maximum margin of 0.3 (refer to Equation. 1 in main text).

**Optimum $\rho$ value**: Table 1 compiles the $\rho$ value used by SAM across various methods on imbalanced CIFAR-10 and CIFAR-100 datasets. The $\rho$ value in these runs is kept constant throughout the duration of training. We adopt a common step $\rho$ schedule for the SAM runs on both ImageNet-LT and iNaturalist 2018. We initialise the $\rho$ with 0.05 for the initial 5 epochs and change it to 0.1 till the 60th epoch. Following that, we increase the $\rho$ value to 0.5 for the final 30 epochs.

**How to select $\rho$ ?** $\rho$ is an hyperparameter in the SAM algorithm and it is important to choose the right value of $\rho$ for best performance on long-tailed learning. We observe that default value of $\rho$ (0.05) as suggested in Foret et al. [7] does not lead to significant gain in accuracy (Refer Fig. 4 in main paper), as it is not able to escape the region of negative curvature. On long-tail CIFAR-10 and CIFAR-100 setting with re-weighting (DRW), a large value of $\rho$ (0.5 or 0.8) seems to work best instead, as in this work our objective to escape saddle points instead of improving generalization. This can be intuitively understood as large regularization ($\rho$) is required for highly imbalanced datasets to escape saddle points as suggested by Theorem 2. In Table 1, we have reported the $\rho$ value used in every experiment. For the large scale datasets like ImageNet-LT and iNaturalist 18, we found that progressively increasing the $\rho$ value gives the best results. This is based on the idea that, as the training progresses, more flatter regions can be recovered from the loss landscape [2]. In our experiments on ImageNet-LT, we use a large $\rho$ of 0.5 in the last 30 epochs of training and we observe

Table 2: Results on ImageNet-LT (ResNet-50) with LDAM+DRW and comparison with other methods. The numbers for methods marked with † are taken from [24].

|  | Two stage | Acc | Head | Mid | Tail |
|---|---|---|---|---|---|
| CE | ✗ | 42.7 | 62.5 | 36.6 | 12.5 |
| cRT [13] † | ✓ | 50.3 | 62.5 | 47.4 | 29.5 |
| LWS [13] † | ✓ | 51.2 | 61.8 | 48.6 | 33.5 |
| MisLAS [24] | ✓ | 52.7 | 61.7 | 51.3 | **35.8** |
| DisAlign [23] | ✓ | 52.9 | 61.3 | **52.2** | 31.4 |
| DRO-LT* [19] | ✗ | **53.5** | **64.0** | 49.8 | 33.1 |
| LDAM + DRW | ✗ | 49.9 | 61.1 | 48.2 | 28.3 |
| LDAM + DRW + SAM | ✗ | 53.1 | 62.0 | 52.1 | 34.8 |

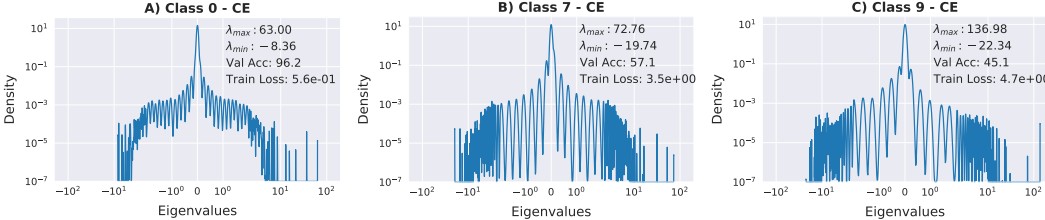

Figure 1: Eigen Spectral Density of Head (Class 0) and Tail (Class 7 and Class 9) with standard CE (without re-weighting). Since CE minimizes the average loss, it can be seen that the loss on the tail class samples (B and C) is quite high. On the head class (A), the loss is low and $\lambda_{min}$ is close to 0.

that the tail accuracy significantly increases at this stage of training. For using the proposed method on a new imbalanced dataset, we suggest starting with $\rho = 0.05$ and increasing $\rho$ till the overall accuracy starts to decrease.

**LPF-SGD and PGD**: We use the official implementation of LPF-SGD [2] [2] to report the results on CIFAR-10 LT and CIFAR-100 LT. For LPF-SGD, we use Monte Carlo iterations $(M) = 8$ and a constant filter radius $(\gamma)$ of 0.001 (as defined in Algorithm 4.1 in Bisla et al. [2]). We implement the stochastic PGD method [11, 12] on our own since there is no official PyTorch implementation available. We sample the perturbation (noise) from a Gaussian distribution with zero mean and $(\sigma)$ standard deviation. We use a $\sigma$ of 0.0001 for CIFAR-10 and CIFAR-100 LT experiments.

**Hessian Experiments**: For calculating the Eigen Spectral Density, we use the PyHessian library [22]. PyHessian uses Lanczos algorithm for fast and efficient computation of the complete Hessian eigenvalue density. The Hessian is calculated on the average loss of the training samples as done in [8, 22]. $\lambda_{min}$ and $\lambda_{max}$ are extracted from the complete Hessian eigenvalue density. It has been shown that the estimated spectral density calculated with the Lanczos algorithm can be used as an approximate to the exact spectral density [8]. Several works [7, 8, 9, 22] have used the same method to calculate spectral density and analyze the loss landscape of neural networks.

All of our implementations are based on PyTorch [18]. For experiments pertaining to imbalanced CIFAR, we use NVIDIA GeForce RTX 2080 Ti, whereas for the large scale ImageNet-LT and iNaturalist 2018, we use NVIDIA A100 GPUs. We log all our experiments with Wandb [1].

## D  Additional Eigen Spectral Density Plots

We find that the spectral density of a class is representative of the other classes in same category (Head, Mid or Tail), hence for brevity we only display the eigen spectrum of one class per category for analysis.

---

[2]https://github.com/devansh20la/LPF-SGD

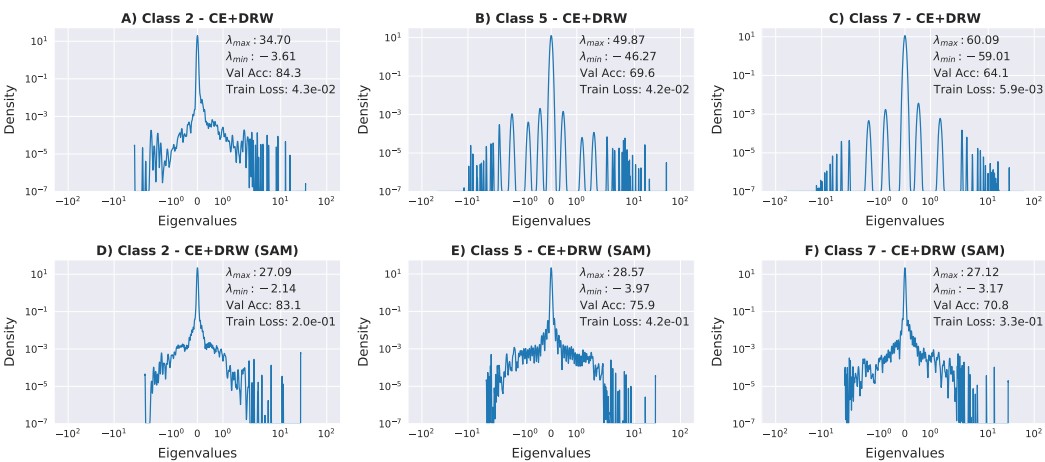

Figure 2: Eigen Spectral Density of the Head (Class 2), Mid (Class 5) and Tail classes (Class 7) with CE+DRW and CE+DRW+SAM.

**CE**: The spectral density on the standard CE loss (without re-weighting) can be seen in Fig. 1. We notice that the density and magnitude of negative eigenvalues is much larger for the tail class (Class 7 and Class 9 in Fig. 1B and 1C) compared to the head classes (Fig. 1A). On the other hand, the spectral density of the head class (Class 0) is very different from that of the tail class, with $\lambda_{min}$ of the head class very close to 0 indicating convergence to minima.

It must be noted that without re-weighting, the loss on the tail class samples is high because CE minimizes the average loss. Hence, the solution may not converge for tail class loss. However, in CE+DRW after re-weighting, we observe that the loss on tail class samples is very low, which indicates convergence to a stationary point. Thus, in CE+DRW, we can evidently conclude that the presence of large negative curvature indicates convergence to a saddle point. In summary, we find that just using CE converges to a point with significant negative curvature in tail class loss landscape. Further, though DRW is able to decrease the loss on tail classes, it still does converge to a point with significant negative curvature. This indicates that it converges to a saddle point instead of a minima. Hence, both CE and CE+DRW do not converge to local minima in tail class loss landscape.

**CE+DRW**: We show additional class wise Eigen Spectral Density plots with CE+DRW and CE+DRW with SAM in Fig. 2. We analyze the spectral density plots on Head (Class 2), Mid (Class 5) and Tail (Class 7). It can be seen that the magnitude of $\lambda_{max}$ and $\lambda_{min}$ is much lower with SAM in all the classes (Fig. 2 D, E, F). This indicates that SAM reaches a flatter local minima with no significant presence of negative eigenvalues, escaping saddle points.

**LDAM**: We also show Spectral density plots of Class 0 (Fig. 3 A, C) and Class 9 (Fig. 3 B, D) with LDAM+DRW method (SGD and SAM) in Fig. 3. The existence of negative eigenvalues in the tail class spectral density (Fig. 3B) indicates that even for LDAM loss (a regularized margin based loss), the solutions do converge to a saddle point. This also indicates that observations with CE+DRW hold good for long-tailed learning methods like LDAM which use margins instead of re-weighting directly. Hence, this gives evidence of the reason why SAM can be combined easily with LDAM, VS Loss etc. to effectively improve performance.

The spectral density of the tail class of LDAM with SAM (Fig. 3D) contains fewer negative eigenvalues compared to SGD (Fig. 3B). This indicates convergence to local minima and clearly explains why SAM improves the performance of LDAM by 12.7%.

## E    Additional Results

For further establishing the generality of our method, we choose two recent orthogonal method Influence-Balanced Loss [17] (IB-Loss) and Parametric Contrastive Learning (PaCo) [5] and apply

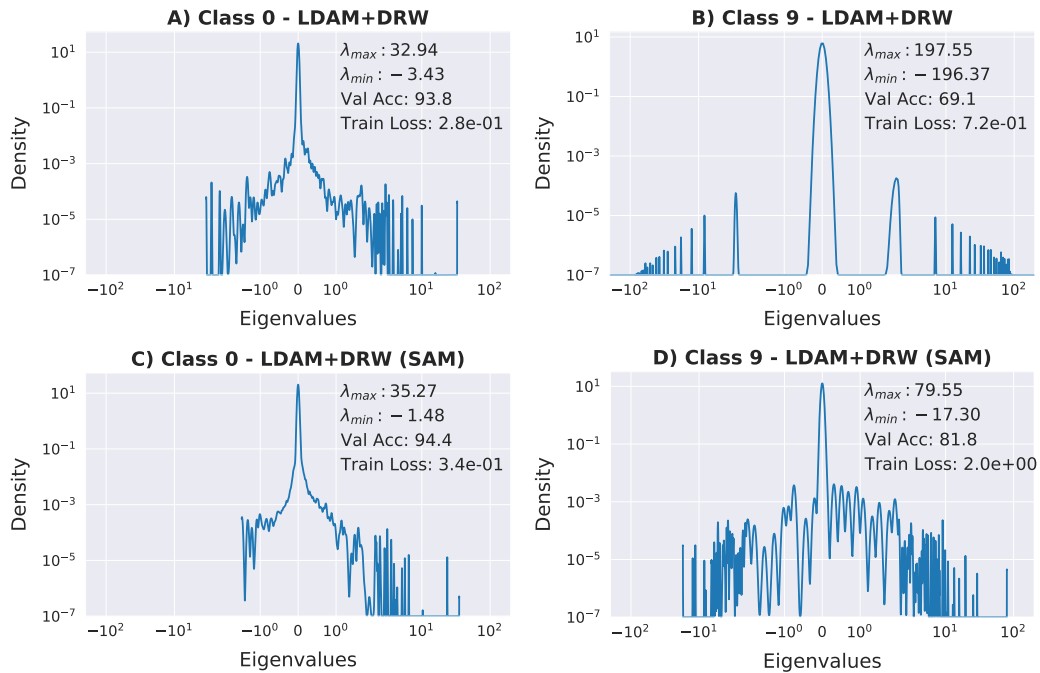

Figure 3: Eigen Spectral Density of the Head (Class 0) and Tail (Class 9) class trained with LDAM. Even with LDAM, we observe existence of negative eigenvalues in the loss landscape for the tail class, which reduce in magnitude when LDAM is used with SAM.

proposed high $\rho$ SAM over them. We use the open-source implementations of IB-Loss [3] and PaCo [4] to reproduce the results and add our proposed method (high $\rho$ SAM) to that setup to obtain the results reported in the table below. We show results on CIFAR-100 LT with an imbalance factor ($\beta$) of 100 and 200. We observe that SAM with high $\rho$ significantly improves overall performance along with the performance on tail classes with both IB-Loss and PaCo method (Table 5). Despite PaCo baseline achieving close to state-of-the-art performance, the addition of high $\rho$ SAM is able to further improve the accuracy. This indicates the generality and applicability of proposed method across various long-tailed learning algorithms.

We show additional results on the large scale ImageNet-LT (Table 2) and iNaturalist 2018 (Table 3 in main text) dataset with LDAM-DRW. We also compare with recent long-tail learning methods: cRT [13], MisLAS [24], DisAlign [23] and DRO-LT [19]. On ImageNet-LT, LDAM+DRW with SAM leads to a 3.2% gain in overall accuracy with 6.5% increase in tail class accuracy. It can be seen that LDAM+DRW+SAM outperforms most other methods, including MisLAS which uses mixup. Also, it is important to note that MisLAS is trained for 180 epochs unlike LDAM+DRW which is trained only for 90 epochs. We observe that LDAM+DRW+SAM surpasses the performance of two-stage training methods including MisLAS, cRT, LWS, and DisAlign. Compared to these two-stage methods, our method is a single stage method and outperforms these two-stage methods. We want to add that we were not able to reproduce the numbers reported in DRO-LT* [19] when we were trying to incorporate SAM with DRO-LT.

With LDAM+DRW, the addition of SAM results in an increase in Head, Mid and Tail categories on iNaturalist 2018 (Table 3 in main paper). Specifically, LDAM+DRW+SAM outperforms all other methods in the tail class accuracy.

This further emphasizes that our analysis is applicable to large scale imbalanced datasets like ImageNet-LT and iNaturalist 2018. We also want to highlight that our analysis shows that high $\rho$ SAM with re-weighting can be used as a *strong baseline* in long tailed visual recognition problem.

---

[3]https://github.com/pseulki/IB-Loss
[4]https://github.com/dvlab-research/Parametric-Contrastive-Learning

Table 3: Results on CIFAR-10 LT with different Imbalance Factor ($\beta$).

| | $\beta = 10$ | | | | $\beta = 50$ | | | |
|---|---|---|---|---|---|---|---|---|
| | Acc | Head | Mid | Tail | Acc | Head | Mid | Tail |
| CE + DRW [3] | 88.3 | 93.6 | 85.3 | 86.9 | 79.9 | 92.2 | 76.5 | 72.0 |
| CE + DRW + SAM | 89.7 | 93.4 | 86.1 | 90.8 | 83.8 | 91.3 | 80.5 | 80.8 |
| LDAM + DRW [3] | 87.8 | 91.9 | 85.0 | 87.5 | 82.0 | 90.9 | 78.7 | 77.5 |
| LDAM + DRW + SAM | 89.4 | 93.4 | 86.2 | 89.8 | 84.8 | 92.8 | 82.1 | 80.4 |
| | $\beta = 100$ | | | | $\beta = 200$ | | | |
| | Acc | Head | Mid | Tail | Acc | Head | Mid | Tail |
| CE + DRW [3] | 75.5 | 91.6 | 74.1 | 61.4 | 69.9 | 91.1 | 70.0 | 48.4 |
| CE + DRW + SAM | 80.6 | 91.4 | 78.0 | 73.1 | 76.6 | 91.5 | 74.9 | 64.0 |
| LDAM + DRW [3] | 77.5 | 91.1 | 75.7 | 66.4 | 72.5 | 90.2 | 72.3 | 54.9 |
| LDAM + DRW + SAM | 81.9 | 91.0 | 79.2 | 76.4 | 78.1 | 91.2 | 75.6 | 68.4 |

Table 4: Results on CIFAR-100 LT with different Imbalance Factor ($\beta$).

| | $\beta = 10$ | | | | $\beta = 50$ | | | |
|---|---|---|---|---|---|---|---|---|
| | Acc | Head | Mid | Tail | Acc | Head | Mid | Tail |
| CE + DRW [3] | 58.1 | 65.6 | 58.5 | 48.2 | 46.5 | 63.3 | 47.5 | 24.4 |
| CE + DRW + SAM | 60.7 | 66.0 | 60.5 | 54.4 | 50.0 | 61.9 | 50.9 | 33.7 |
| LDAM + DRW [3] | 57.8 | 67.5 | 58.9 | 44.5 | 47.1 | 62.9 | 48.2 | 26.1 |
| LDAM + DRW + SAM | 60.1 | 70.2 | 61.3 | 46.1 | 49.4 | 66.1 | 50.2 | 27.8 |
| | $\beta = 100$ | | | | $\beta = 200$ | | | |
| | Acc | Head | Mid | Tail | Acc | Head | Mid | Tail |
| CE + DRW [3] | 41.0 | 61.3 | 41.7 | 14.7 | 36.9 | 59.7 | 36.1 | 9.6 |
| CE + DRW + SAM | 44.6 | 61.2 | 47.5 | 20.7 | 41.7 | 63.4 | 43.0 | 13.1 |
| LDAM + DRW [3] | 42.7 | 61.8 | 42.2 | 19.4 | 38.3 | 58.8 | 36.3 | 15.1 |
| LDAM + DRW + SAM | 45.4 | 64.4 | 46.2 | 20.8 | 42.0 | 63.0 | 41.4 | 16.6 |

We also find that SAM is highly compatible with different loss-based methods (like LDAM, VS) for tackling imbalance and can be used to achieve significantly better performance.

## F  Additional Results with Varying Imbalance Factor

We show the results with different imbalance factors ($\beta = 10, 50, 100$ and $200$) on CIFAR-10 LT (Table 3) and CIFAR-100 LT (Table 4) datasets with two methods. It can be seen that the observations in Table 1 are applicable with different degrees of imbalance. SAM with re-weighting improves upon the performance of CE and LDAM losses in all the experiments with varied imbalance factor. We observe an average increase of 3.9% and 3.2% on CIFAR-10 LT and CIFAR-100 LT datasets, respectively. This gain in performance is primarily due to the improvement in the tail accuracy, which increases by 8.6% on CIFAR-10 LT and 3.9% on CIFAR-100 LT.

As the dataset becomes more imbalanced ($\beta$ increases), the gain in accuracy with SAM on the tail classes improves significantly. For instance, on CIFAR-10 LT with $\beta = 10$ (Table 3), CE+DRW+SAM improves upon CE+DRW by 1.2% with a 3.9% increase in tail class accuracy. However, with a more imbalanced dataset (*i.e.* CIFAR-10 LT $\beta = 200$), SAM leads to a 6.7% boost in overall accuracy with a massive 15.6% increase in the tail class performance.

Table 5: Results on CIFAR-100 LT with IB-Loss and PaCo.

|  | $\beta = 100$ | | $\beta = 200$ | |
|---|---|---|---|---|
|  | Acc | Tail | Acc | Tail |
| IB [3] | 40.4 | 14.9 | 36.7 | 10.3 |
| IB + SAM | 42.8 | 25.0 | 37.7 | 17.8 |
| PaCo [5] | 51.5 | 33.9 | 47.0 | 26.9 |
| PaCo + SAM | 53.0 | 36.0 | 48.0 | 27.8 |

---

**Algorithm 1** DRW + SAM

---

**Require:** Network $g$ with parameters $w$; Training set $\mathbb{S}$; Batch size $b$; Learning rate $\eta > 0$; Neighborhood size $\rho > 0$, Neighborhood size for re-weighted loss $\rho_{drw} >= \rho$; Total Number of Iterations $E$; Deferred Reweighting Threshold $T$; Number of samples in class $y$: $n_y$; Loss Function $\mathcal{L}$ (Cross-Entropy, LDAM).

1: **for** $i = 1$ to $E$ **do**
2:      Sample a mini-batch $\mathbb{B} \subset \mathbb{S}$ with size $b$.
3:      **if** $E < T$ **then**
4:          Compute Loss $\mathcal{L} \leftarrow \frac{1}{b} \sum_{(x,y) \in \mathbb{B}} \mathcal{L}(y; g_w(x))$
5:          Compute $\epsilon \leftarrow \rho * \nabla_w \mathcal{L} / ||\nabla_w \mathcal{L}||$         ▷ Compute Sharp-Maximal Point
6:          Compute Loss at $w + \epsilon$; $\mathcal{L} \leftarrow \frac{1}{b} \sum_{(x,y) \in \mathbb{B}} \mathcal{L}(y; g_{w+\epsilon}(x))$
7:          Calculate gradient $d$: $d \leftarrow \nabla_w \mathcal{L}$
8:      **else**         ▷ Deferred Re-Weighting (DRW)
9:          Compute re-weighted Loss $\mathcal{L}_{\text{RW}} \leftarrow \frac{1}{b} \sum_{(x,y) \in \mathbb{B}} n_y^{-1} \cdot \mathcal{L}(y; g_w(x))$
10:          Compute $\epsilon \leftarrow \rho_{drw} * \nabla_w \mathcal{L}_{\text{RW}} / ||\nabla_w \mathcal{L}_{\text{RW}}||$
11:          Compute re-weighted Loss at $w + \epsilon$; $\mathcal{L}_{\text{RW}} \leftarrow \frac{1}{b} \sum_{(x,y) \in \mathbb{B}} n_y^{-1} \cdot \mathcal{L}(y; g_{w+\epsilon}(x))$
12:          Calculate gradient $d$: $d \leftarrow \nabla_w \mathcal{L}_{\text{RW}}$
13:      Update weights $w_{i+1} \leftarrow w_i - \eta d$

---

# G   Algorithm

We describe our method in detail in Algorithm 1. On the large scale ImageNet-LT and iNaturalist-18 dataset, we use $\rho_{drw} > \rho$. For CIFAR-10 LT and CIFAR-100 LT, we find that $\rho = \rho_{drw}$ works well.

# H   Related Work: Long-tailed Learning

In this section, we discuss some recent approaches in long-tailed learning. Equalization loss is proposed in Tan et al. [20] based on the proposition that the gradients of negative samples overpower the gradient of positive samples for minority classes. Influence-Balanced Loss [17] is a sample-level re-weighting method that reweights each sample by the inverse of the norm of the gradient of each sample. The gradient of each sample estimates the influence of that sample in determining the decision boundary. Distill the Virtual Examples (DiVE) [10] addresses the problem of class-imbalanced learning from the lens of knowledge distillation. It is shown that the teacher models' predictions (virtual examples) can be distilled into the student model by making use of cross-category interactions. This leads to an improvement in the accuracy of the minority class samples.

Self-Supervised Learning methods have been shown to learn generalizable representations [4] which are useful for a wide variety of downstream tasks. Self-Supervised pre-training (SSP) has been shown to improve the performance of class-imbalanced learning [21]. Parametric Contrastive Learning (PaCo) [5] introduces parametric class-wise learnable centers into the Supervised Contrastive Learning [14] framework to improve the performance on imbalanced datasets. PaCo achieves close to state-of-the-art performance on most of the long-tailed learning benchmarks. Self Supervised to Distillation (SSD) [16] is a multi-stage training framework for long-tailed recognition with a total of four stages of training. The first two stages involve self-supervised training followed by the generation of soft labels. The final two stages include joint training with distillation and classifier fine-tuning. Balanced Contrastive Learning (BCL) [25] adapts the Supervised Contrastive framework [14] by proposing a

Balanced Contrastive loss which ensures that the feature space is balanced when training with an imbalanced dataset.

# I  Code and License Details

Our codebase is derived from the official implementation of LDAM-DRW[3][5], VS-Loss [15][6] and SAM[7][7] which have been released under the `MIT` license. We have included the code and the pretrained weights of the CE+DRW model trained of CIFAR-10 LT in the supplementary material. The code to reproduce the experiments is available at https://github.com/val-iisc/Saddle-LongTail.