# OpenReview forum: "Escaping Saddle Points for Effective Generalization on Class-Imbalanced Data"
_NeurIPS.cc/2022/Conference — NeurIPS 2022 Accept_

### Official Review · Reviewer_fv7S · 2022-07-07

**Rating:** 6
**Confidence:** 3
**Soundness:** 4 excellent
**Presentation:** 3 good
**Contribution:** 3 good

**Summary:**

In this paper, the authors find that the key issue for imbalanced classification is that the training for minority classes can lead to convergence to saddle points of their loss landscape. This phenomenon cannot be avoided in the reweighting methods. Consequently, the network suffers from poor generalization on minority classes. To escape from the saddle points, they introduce Sharpness-Aware Minimization (SAM) with a high regularization factor $rho$ to enhance the generalization performance. They theoretically and empirically demonstrate that SAM with high $rho$  is able to escape saddle points faster than SGD and converge to better solutions.

**Questions:**

1. Can you explain why the negative eigenvalues appear on imbalanced datasets and disappear on balanced datasets? Do they appear because of the negative gradients from the majority classes? If so, please analyse the differences and compare with the equalization loss (Equalization loss for long-tailed object recognition, CVPR 2020).
2. Is SAM only applied on the loss of the minority classes? I think providing an overall algorithm is better.

**Limitations:**

Does the saddle points exist in other methods, such as data augmentation and sample-level reweighting methods?

**Strengths And Weaknesses:**

Pros:
1. Analysis the problem of imbalanced classification from the persepective of optimaization peocedure is novel.
2. The conclusion that saddle points cannot be avoided in reweighting methods is interesting.
3. Sufficient theoretically and empirically analysis.

---

> ### Author Response · Authors · 2022-08-03
> **Response to Reviewer fv7S (1/2)**
>
> We thank the reviewer for the positive feedback and interesting questions.
>
> > Can you explain why the negative eigenvalues appear on imbalanced datasets and disappear on balanced datasets?
>
> As neural networks have a high capacity, when the amount of data is limited and they are trained without regularization, they potentially overfit and converge to non-convex solutions, which generalize poorly [1]. Whereas, when neural networks are trained on large datasets like ImageNet, they don't require explicit regularization as the implicit regularization of SGD makes them converge to solutions that generalize well.
>
> Similarly, on imbalanced datasets, a re-weighting term is usually applied to the limited tail class samples to increase their importance artificially. This forces the model to find a point of low loss (i.e., overfitting) in the highly non-convex tail class loss landscape. This leads to convergence to a saddle point in a non-convex region (i.e., negative eigenvalues) with low training loss and poor generalization. In Fig. 3A, it can be seen that due to re-weighting (after 160 epochs), the relative magnitude of negative eigenvalues (i.e., $\lambda_{min}/\lambda_{max}$) increases, and the network tends to overfit to tail classes, which in turn leads to saddle points. Hence, re-weighting and limited tail class samples are the reason for convergence to saddle points.
>
> In the case of balanced datasets, there is abundant data and no requirement of re-weighting; hence the implicit regularization of SGD itself is able to help escape saddle points and converge to minima. Hence, negative eigenvalues do not appear for these cases [2].
>
> [1] Poggio, Tomaso, et al. "Theory of deep learning iii: the non-overfitting puzzle." CBMM Memo 73 (2018).
> [2] An Investigation into Neural Net Optimization via Hessian Eigenvalue Density
>
> > Do they appear because of the negative gradients from the majority classes? If so, please analyse the differences and compare with the equalization loss (Equalization loss for long-tailed object recognition, CVPR 2020).
>
> The Equalization loss paper considers each positive sample of one category as a negative sample for other categories. The negative gradients in the Equalization loss paper refer to the gradient of the negative samples, which are larger in magnitude for tail classes compared to positive gradients. However, we focus on Re-Weighted loss functions in our work where we use a factor of $1/n_y$ (i.e., the inverse of the frequency of class y) to re-weight the loss of each sample. In Appendix J, we theoretically show that for such a loss, the (expectation of) gradient norm is equal for the head and tail classes. The negative eigenvalues discussed in our paper are computed on the Hessian of the loss, which are second-order properties in comparison to the first-order property of the negative gradient. Hence, we believe that the appearance of negative eigenvalues is orthogonal to the effect of negative gradients.
>
> > Is SAM only applied on the loss of the minority classes? I think providing an overall algorithm is better.
>
> We apply SAM on overall loss throughout the training, with the major gains being obtained when SAM is applied on the re-weighted loss in the deferred re-weighting (DRW) setting. Re-weighting ensures that the loss of minority classes is enhanced, which implicitly ensures that SAM has a higher contribution from the tail classes. Also, we would like to mention that a higher $\rho\_{DRW}$ is beneficial in the re-weighting phase. We find that using SAM with re-weighted loss leads to a huge gain in tail accuracy in our experiments, whereas just using SAM with overall loss (without re-weighting) leads to sub-optimal performance on tail classes (Table-1,2). We have added the algorithm in Appendix Section I.

---

> > ### Author Response · Authors · 2022-08-03
> > **Response to Reviewer fv7S (2/2)**
> >
> > > Does the saddle points exist in other methods, such as data augmentation and sample-level reweighting methods?
> >
> > Thank you for this interesting question. We believe that methods that involve re-weighting (including sample-level re-weighting) to increase focus on tail class samples can suffer from convergence to non-convex regions (possibly saddle points) for tail classes. To give some evidence into this, we run IB (influence-balanced) loss, a sample-level reweighting method, and apply high $\rho$ SAM on it. We find that it can effectively improve overall performance by majorly improving performance on tail classes. (Kindly refer to the General Response for the results).
> >
> > Further, we also observe an increase in performance when the proposed method is applied to Parametric Contrastive Learning (a contrastive method with strong data augmentations). Hence, we believe that data augmentation may help limit the presence of saddle points but cannot prevent it entirely. Hence our method should still be effective in such cases. This is also evidenced in literature as SAM with strong augmentation can still provide effective improvement for various tasks under limited data [1]. The analysis of the loss landscape is non-trivial in the case of methods with multiple loss components and sample-level reweighting, hence characterizing the obtained solution as a saddle point is a challenging task for future work.
> > ***
> > [1] Chen, Xiangning, Cho-Jui Hsieh, and Boqing Gong. "When Vision Transformers Outperform ResNets without Pre-training or Strong Data Augmentations." International Conference on Learning Representations. 2021.

---

### Official Review · Reviewer_hF5j · 2022-07-11

**Rating:** 6
**Confidence:** 2
**Soundness:** 3 good
**Presentation:** 3 good
**Contribution:** 3 good

**Summary:**

The authors find that for the tail class loss landscape, the solution converges to a saddle point and the network thus suffers from poor generalization on minority classes. This work uses Sharpness-Aware Minimization (SAM) to escape saddle points and enhance the generalization performance, which has been theoretically and empirically demonstrated. Experimental results show that combining SAM with SOTA techniques leads to significant gains in performance on minority classes.

**Questions:**

How the sharp maximal point ϵ and weight w are updated in this work?

What is the difference between  θ and w?

**Ethics Review Area:**

["Inadequate Data and Algorithm Evaluation", "I don’t know"]

**Limitations:**

1 The Algorithm should be presented and described in detail, which is helpful for understanding the proposed method.
2 The background of Sharpness-Aware Minimization (SAM) shoud be described in detail.


**Strengths And Weaknesses:**

Strength
1 The idea of studying the effect of negative curvature for class-imbalanced learning using deep neural networks is novel and interesting, which may pave a new way for further research on imbalanced classification.
2  Experimental results empirically show that escaping saddle points with SAM leads to a notable in crease in overall accuracy primarily due to the major gain in the accuracy on the tail classes.

weakness
1 The Algorithm should be presented and described in detail.
2 The background of Sharpness-Aware Minimization (SAM) shoud be described in detail.

---

> ### Author Response · Authors · 2022-08-02
> **Response to Reviewer hF5j**
>
> We thank the reviewer for providing encouraging comments on our paper.
>
> > The Algorithm should be presented and described in detail.
>
> Thanks for your suggestion, we have added the Algorithm in Appendix I (present in Supplementary material) of the revised version.
>
> > The background of Sharpness-Aware Minimization (SAM) shoud be described in detail.
>
> Several works [1,2] have shown that flat minima generalizes better than sharp minima.
> Sharpness-Aware Minimization is a method that aims to explicitly reach a flat minima solution. The goal of SAM is not only to reach a minima with a low loss but also to ensure that the neighbourhood of the minima has a low loss, i.e. convergence to a plateau region. SAM formulates this as a min-max optimization problem.
>
> $$    \underset{w}{\min} \;\underset{||\epsilon|| \leq \rho}{\max}\; f(w + \epsilon) $$
>
> In the first step, SAM tries to find the region in the neighourbood with the maximum loss (sharp maximal point).
>
> $$  \hat{\epsilon}(w) \approx \underset{||\epsilon|| \leq \rho}{\arg \max} \;  f(w) + \epsilon^T\nabla f(w) \;
>             = \rho \nabla f(w) / ||\nabla f(w)||\_2 $$
>
> And, in the second step, SAM tries to minimize loss on this high-loss region using gradient descent.
>
> $$\underset{w}{\min}  f(w + \hat{\epsilon}(w)) $$
>
> It has been shown that SAM generalizes better than its SGD counterparts. SAM has also been shown to be applicable in a wide variety of tasks from supervised classification [3] to meta-learning [4] to Natural Language Tasks [5]. Several variants of SAM [6, 7, 8] have been proposed which improve upon the performance of SAM. We have improved SAM description and will also provide the background of SAM in detail in the final version.
>
>
>
> > How the sharp maximal point ϵ and weight w are updated in this work?
>
> The sharp maximal point $\epsilon$ is added to the weights of the model ( $w + \epsilon$ ). The loss is now calculated at $w + \epsilon$ and the weights are updated from $w$. We hope that this clarifies your doubt. We also refer you to the Algorithm (Appendix I) where this has been described.
>
> > What is the difference between θ and w?
>
> $\theta$ and $w$ are used interchangebly. We thank the reviewer for pointing this out. We have now made the notation consistent in the main paper. (L178-188)
>
> Please let us know in case you require any further clarifications.
> ***
> [1] Sepp Hochreiter and Jürgen Schmidhuber. Flat minima. Neural computation, 9(1):1–42, 1997. \
> [2] Nitish Shirish Keskar, Dheevatsa Mudigere, Jorge Nocedal, Mikhail Smelyanskiy, and Ping Tak Peter Tang. On large-batch training for deep learning: Generalization gap and sharp minima. arXiv preprint arXiv:1609.04836, 2016. \
> [3] Pierre Foret, Ariel Kleiner, Hossein Mobahi, and Behnam Neyshabur. Sharpness-aware minimization for efficiently improving generalization. In International Conference on Learning Representations, 2021. URL https://openreview.net/forum?id=6Tm1mposlrM.  \
> [4] Abbas, M., Xiao, Q., Chen, L., Chen, P. Y., & Chen, T. (2022). Sharp-MAML: Sharpness-Aware Model-Agnostic Meta Learning. arXiv preprint arXiv:2206.03996. \
> [5] Bahri, D., Mobahi, H., & Tay, Y. (2021). Sharpness-aware minimization improves language model generalization. arXiv preprint arXiv:2110.08529. \
> [6] Du, Jiawei, et al. "Efficient sharpness-aware minimization for improved training of neural networks." arXiv preprint arXiv:2110.03141 (2021). \
> [7] Kwon, J., Kim, J., Park, H., & Choi, I. K. (2021, July). Asam: Adaptive sharpness-aware minimization for scale-invariant learning of deep neural networks. In International Conference on Machine Learning (pp. 5905-5914). PMLR. \
> [8] Zhao, Yang, Hao Zhang, and Xiuyuan Hu. "SS-SAM: Stochastic Scheduled Sharpness-Aware Minimization for Efficiently Training Deep Neural Networks." arXiv preprint arXiv:2203.09962 (2022).

---

> > ### Comment · Reviewer_hF5j · 2022-08-09
> > **Thanks for your reply.**
> >
> > I appreciate your response, where most of my concerns are resolved.

---

### Official Review · Reviewer_s5kU · 2022-07-11

**Rating:** 6
**Confidence:** 3
**Soundness:** 3 good
**Presentation:** 3 good
**Contribution:** 3 good

**Summary:**

In the real-world, many datasets are imbalanced, which can degrade the trained model’s performance. To tackle this problem, this paper analyzes the class-imbalance problem by examining the spectral density of Hessian of class-wise loss. From the observation that the tail class loss converges to a saddle point, this paper proposes to use sharpness aware minimization (SAM). This paper justifies the proposed method theoretically and empirically on multiple benchmark datasets.

**Questions:**

Please refer to the weakness.

**Limitations:**

The limitations and potential negative societal impact are addressed,

**Strengths And Weaknesses:**

Strengths
- This paper justifies the proposed method with theoretical justification and various analyses.
- The proposed method improves the baselines.
- The paper is well written.

Weaknesses
- The comparison baselines and the prior literature on long-tailed learning is limited.

---

> ### Author Response · Authors · 2022-08-02
> **Response to Reviewer s5kU**
>
> We thank the reviewer for the feedback and positive comments on our work.
>
> > The comparison baselines and the prior literature on long-tailed learning is limited.
>
> We primarily analyze the SOTA loss manipulation techniques like Label Distribution Aware Margin Loss (LDAM) and Vector Scaling (VS), as we analyze the models trained on long-tailed distribution through the lens of loss landscape. We do not analyse some sophisticated techniques [1,2] as they use multiple loss functions along with using multiple networks (experts), complex data augmentations etc. which complicates loss landscape analysis. However, we would like to convey that even all these sophisticated techniques *use some kind of reweighting (loss manipulation)* (eg. LDAM in [2]) to improve performance on tail classes. Hence, we believe that our proposed method of high $\rho$ SAM can also help in these cases.
>
> For further enforcing our claim, we have now shown the effectiveness of the proposed method with Influence-Balanced Loss [3] (IB-Loss) and Parametric Contrastive Learning [4] (PaCo) which utlizies data augmentation and self-supervision. It can be seen that SAM significantly improves the tail class performance with both IB-Loss and PaCo on CIFAR-LT datasets. (Kindly refer to the general response)
>
> Additionally, we have added a section on the prior literature in long-tailed learning in Appendix K. We have discussed about recent long-tailed learning based methods including PaCo and IB-Loss. We will add these additional comparisons and prior literature in the the final version.
> ***
>
> [1] Long-tailed recognition by routing diverse distribution-aware experts. In International Conference on Learning Representations, 2021. \
> [2] Test-agnostic long-tailed recognition by test-time aggregating diverse experts with self-supervision, 2021. \
> [3] Influence-Balanced Loss for Imbalanced Visual Classification, ICCV 2021. \
> [4] Parametric Contrastive Learning, ICCV 2021.

---

> > ### Comment · Reviewer_s5kU · 2022-08-09
> > **Thank you for the response.**
> >
> > Thank you for the response.
> > I hope the authors to add the comparison results and other reviewers' responses in the final version.

---

### Official Review · Reviewer_oRk4 · 2022-07-12

**Rating:** 6
**Confidence:** 3
**Soundness:** 4 excellent
**Presentation:** 4 excellent
**Contribution:** 2 fair

**Summary:**

The authors analyze the class-imbalanced learning problem through the lens of the loss landscape.
Specifically, they examine the spectral density of Hessian of class-wise loss, in which they observe that for the tail class loss landscape, the solution converges to a saddle point.
They further theoretically and empirically demonstrate that Sharpness Aware Minimization (SAM), a recent technique that aims to converge to flat minima, can be effectively used to escape saddle points for minority classes.
Borrowing SAM results in an increase in accuracy in the minority classes.


**Questions:**



**Limitations:**

The authors addressed the limitations of the proposed method and the potential negative social impact of their work.

**Strengths And Weaknesses:**

Strength

- The idea to interpret the long-tailed distribution problem with respect to the loss of landscape is novel.
- The authors validate the effect of the SAM for model training theoretically and visually.

Weakness

The proposed solution for the long-tailed recognition problem is not novel enough. It would be better to propose a novel method by modifying the existing SAM method to be especially helpful for the tail classes. Simply borrowing the existing method, which is generally helpful, might not be enough for complete research. Also, empirically in Table 1,2,3, it is hard to see the SAM is especially helpful for the tail classes (especially helpful to alleviate the long-tailed distribution).

The experiments also seem to be weak. For example, in Table1, the authors should also evaluate their method with other imbalance ratios.

Also, they should compare with more recent state-of-the-art long-tailed recognition works [1,2,3,4]; the baselines shown in the paper are too weak. In order to claim the effectiveness of the proposed method, the authors should apply the proposed method to the state-of-the-art methods and see the improvements instead of only applying the method on simple baselines.

[1] Self Supervision to Distillation for Long-Tailed Visual Recognition, ICCV 2021
[2] Parametric Contrastive Learning, ICCV 2021
[3] Influence-Balanced Loss for Imbalanced Visual Classification, ICCV 2021
[4] Distilling Virtual Examples for Long-tailed Recognition, ICCV 2021

========== ------- Comments after the rebuttal ------========
I have read the other reviewer's comments and the author's rebuttal, which addresses most of my concerns. Therefore, I would like to raise my score.

---

> ### Author Response · Authors · 2022-08-03
> **Response to Reviewer oRk4 (1/2)**
>
> We thank the reviewer for the detailed feedback on our paper. We provide clarifications to the concerns below:
>
> > The proposed solution for the long-tailed recognition problem is not novel enough. Simply borrowing the existing method, which is generally helpful, might not be enough for complete research.
>
> We would like to highlight that one of the main contributions of our work is to *show that with re-weighting based methods, the model converges to a saddle point in the tail class* loss landscape. SAM is one of the techniques in addition to others like PGD etc. (See Table 4), which we find to be effective in escaping saddle points and can be used for enhancing performance significantly on tail classes.
>
> Also, just using SAM does not improve performance (as you observed correctly in rows 1,2 in Table 1,2). It is the *proposed effective combination of re-weighting (DRW) + SAM* that leads to escape from saddle points and improvement in tail class performance (Table-1,2,3).
>
> The performance improvement due to the use of **SAM with high $\rho$ value** comes primarily from escaping saddle points, which is in contrast to earlier works that improve accuracy by reaching a flat minima with SAM. The effectiveness of SAM for avoiding saddle points is a **novel property** that we discover in this work.
>
> This is the first work to analyze the long-tailed visual recognition problem through the lens of loss landscape and the dynamics of optimization, to the best of our knowledge. We believe that this is an interesting direction which can lead to significant new work in long-tailed learning.
>
> > Also, empirically in Table 1,2,3, it is hard to see the SAM is especially helpful for the tail classes
>
> We would like to clarify that SAM in itself is not effective for tail classes (row 1-2), however our proposed combinations of *SAM + Reweighting (eg. VS, DRW etc.)* significantly improves the performance by an average of 4.1% in Table 1 and 3.9% in Table 2. Particularly, the tail class accuracy increases by an average of 6.9% in Table 1 and 3.9% in Table 2. Similar trend is seen in Table 3, which reports the results on large scale benchmarks. SAM with Re-weighting improves the accuracy by 2.1% on ImageNet-LT and 2.3% on iNat18, with performance of tail classes increasing by 6.5% and 3.2% on ImageNet-LT and iNat18, respectively. This clearly demonstrates the effectiveness of SAM with re-weighting based methods.
>
>
> > The experiments also seem to be weak. For example, in Table1, the authors should also evaluate their method with other imbalance ratios.
>
> **Evaluation across Imbalance**
> We provide evaluation of our proposed methods across various imbalance ratios ( $\beta$ ) i.e. (10, 50, 100 and 200) in Table 4 and Table 5 of Appendix (Section H). We find that our method is effective across different imbalance ratios and the performance improvement due to our proposed method increases, as imbalance in dataset increases.  For instance, on CIFAR-10 LT with imbalanced ratio ( $\beta$ = 10) (Table 4 in Appendix), CE+DRW+SAM improves upon CE+DRW by 1.2\% with a 3.9\% increase in tail class accuracy. However, with a more imbalanced dataset (i.e CIFAR-10 LT $\beta$ = 200), SAM leads to a 6.7\% boost in overall accuracy with a massive 15.6\% increase in the tail class performance. We observe an average increase of 3.9% and 3.2% on CIFAR-10 LT and CIFAR-100 LT datasets, respectively. This gain in performance is primarily due to the improvement in the tail accuracy, which increases by 8.6% on CIFAR-10 LT and 3.9% on CIFAR-100 LT. We kindly refer the reviewer to the Appendix H (Table 4 and Table 5) for detailed results and explanation.
>
>
> > Comparison with other state of the art methods
>
> Kindly refer to the General response for the results on CIFAR-100 LT with IB-Loss [1] and PaCo [2]. The improved results with the proposed method indicate the effectiveness of the proposed method with recent state-of-the-art baseline methods.
>
> Further, we would like to mention that majority of methods in literature use some variant of loss re-weighting (i.e. LDAM etc.) to improve performance on tail classes, hence we expect our solution to also *provide improvement* there. We have also compared with recent loss re-weighting method of VS Loss [NeurIPS2021] and shown effective improvement over that in Table-1,2. Long-tailed methods based on contrastive learning or ensemble-based ideas incorporate multiple losses and the analysis of loss landscape is not trivial in that scenario. As we specifically wanted to analyze the effect of re-weighted losses in the loss landscape, we have primarily shown results on the loss manipulation based methods. However, with the improved results on IB-Loss and PaCo, we demonstrate that our method can be used along with a wide variety of methods.

---

> > ### Author Response · Authors · 2022-08-03
> > **Response to Reviewer oRk4 (2/2)**
> >
> > > Additional Related Work
> >
> > We have provided a discussion on the works mentioned in the review in Appendix K.
> >
> > Please let us know in case you require any further clarifications.
> >
> > ***
> > [1] Influence-Balanced Loss for Imbalanced Visual Classification, ICCV 2021 \
> > [2] Parametric Contrastive Learning, ICCV 2021.

---

> > ### Comment · Reviewer_oRk4 · 2022-08-08
> > **Response**
> >
> > Thank you, authors, for your elaborate rebuttal. I have read the other reviewer's comments and the author's rebuttal, which addresses most of my concerns. Therefore, I would like to raise my score to 6.

---

> ### Author Response · Authors · 2022-08-08
> **Request for Response**
>
> Dear Reviewer,
> We sincerely thank you for your time and efforts in reviewing our work. We have provided clarification to the concerns you have mentioned in our response. We have also shown additional results wherever possible to further clear up your doubts. As the author-reviewer discussion period is ending soon, we sincerely request to please let us know if you have any further concerns.
>
> Thanks
> Authors

---

### Author Response · Authors · 2022-08-02
**General Response to All Reviewers**

We thank all the reviewers for their time and valuable feedback.
## Summary of Revision to Paper and Supplementary

### Main Paper
1. We modify the notation in the related work to maintain consistency throughout the paper.
2. We fix the notation of $\theta$ and $w$.

### Supplementary
1. Additional Results with Varying Imbalance Factor (Appendix H): As suggested by reviewer oRk4, we show results on CIFAR-10 LT and CIFAR-100 LT with different imbalance factors.
2. Algorithm (Appendix I): We provide a detailed description of our algorithm. (Reviewers hF5j and fv7S)
3. Analysis of Gradient for Reweighted Loss (Appendix J): We compare with equalization loss. (Reviewer fv7S)
4. Related Work: Long-tailed Learning (Appendix K): We elaborate on the prior literature in long-tailed learning. (Reviewer s5kU and oRk4)

## Additional Results on CIFAR-100 LT

For further establishing the generality of our method, we choose two recent orthogonal method Influence-Balanced Loss[1] (IB-Loss) and Parametric Contrastive Learning (PaCo)[2] and apply proposed high $\rho$ SAM over them. We use the open-source implementations of IB-Loss (https://github.com/pseulki/IB-Loss) and PaCo (https://github.com/dvlab-research/Parametric-Contrastive-Learning) to reproduce the results and add our proposed method (high $\rho$ SAM) to that setup to obtain the results reported in the table below. We show results on CIFAR-100 LT with an imbalance factor ( $\beta$ ) of 100 and 200. We observe that SAM with high $\rho$ signficantly improves overall performance along with the performance on tail classes with both IB-Loss and PaCo method. Despite PaCo baseline achieving close to state-of-the-art performance, the addition of high $\rho$ SAM is able to further improve the accuracy. This indicates the generality and applicability of proposed method across various long-tailed learning algorithms. We will add these baselines and results in the final version.

**Table:** Performance of IB-Loss on CIFAR-100 LT
| Method | Acc | Tail |
| ------ | ------ | ------ |
| IB [1] ( $\beta$-100 ) | 40.4 | 14.9 |
| IB + SAM ( $\beta$-100 ) | 42.8 | 25.0 |
| IB [1] ( $\beta$-200 ) | 36.7 | 10.3 |
| IB + SAM ( $\beta$-200 ) | 37.7| 17.8 |

**Table:** Performance of PaCo on CIFAR-100 LT
| Method | Acc | Tail |
| ------ | ------ | ------ |
| PaCo [2] ( $\beta$-100 ) | 51.5 | 33.9 |
| PaCo + SAM ( $\beta$-100 ) | 53.0 | 36.0 |
| PaCo [2] ( $\beta$-200 ) | 47.0 | 26.9 |
| PaCo + SAM ( $\beta$-200 ) | 48.0 | 27.8 |
***
[1] Influence-Balanced Loss for Imbalanced Visual Classification, ICCV 2021 \
[2] Parametric Contrastive Learning, ICCV 2021

---

### Author Response · Authors · 2022-08-08
**General Request for Response**

We thank all the reviewers for their interesting questions and constructive feedback. We have carefully responded to each point raised in the reviews. We hope that the response clarifies all the questions. Please let us know if any further clarifications are required.

---

### Meta-Review · Area_Chair_L8Cc · 2022-08-22

**Recommendation:** Accept
**Confidence:** Certain

**Metareview:**

The paper studies the problem of saddle point escape for class imbalanced datasets and mostly makes two contributions from my perspective:
1) Analysis of the spectral density of the Hessian for class-imbalanced datasets. This observation is novel as far as I know.
2) A short analysis of SAM demonstrating it escapes saddle points

While the first point seems novel and of interest, I have some limited reservations regarding the second contribution. Theoretically, the authors provide a theorem demonstrating that the CNC condition derived in Daneshmand et al. holds with a larger constant. It is however unclear whether this is the reason for the superior performance of SAM in unbalanced datasets. Saddle points are often not prevalent in the loss landscapes of modern neural networks. The paper does not directly show that better performance is linked to saddles. I would like to encourage the authors to more directly highlight the importance of the CNC condition used in the analysis.

Overall, the reviewers are still rather positive about the paper and despite its shortcomings, it has the potential to encourage more research in this field. I, therefore, recommend acceptance and invite the authors to add a discussion of the shortcomings that should be addressed in future work.

Finally, I note that there is some recent work analyzing the dynamics of gradient descent under class imbalance:
Characterizing the Effect of Class Imbalance on the Learning Dynamics, Francazi et al.
The findings do not seem to be directly related but it's probably worth checking.


**Award:**

No

---

### Decision · Program_Chairs · 2022-09-14

Accept